# Seasonal Variation of Rainy and Dry Season Per Capita Water Consumption in Freetown City Sierra Leone

**Salmatta Ibrahim A \***, **Fayyaz Ali Memon** and **David Butler**

Centre for Water Systems, College of Engineering, Mathematics and Physical Sciences, Harrison Building, North Park Road, University of Exeter, Exeter EX4 4QF, UK; f.a.memon@exeter.ac.uk (F.A.M.); d.butler@exeter.ac.uk (D.B.)

**\*** Correspondence: si255@exeter.ac.uk or salmattaibrahim@gmail.com

**Abstract:** Ensuring a sustainable urban water supply for developing/low-income countries requires an understanding of the factors affecting water consumption and technical evidence of individual consumption which can be used to design an improved water demand projection. This paper compared dry and rainy season water sources available for consumption and the end-use volume by each person in the different income groups. The study used a questionnaire survey to gather household data for a total of 398 households, which was analysed to develop the relationship between per capita water consumption characteristics: Socio-economic status, demographics, water use behaviour around indoor and outdoor water use activities. In the per capita water consumption patterns of Freetown, a seasonal variation was found: In the rainy season, per capita water consumption was found to be about 7% higher than the consumption for the full sample, whilst in the dry season, per capita water consumption was almost 14% lower than the full survey. The statistical analysis of the data shows that the average per capita water consumption for both households increases with income for informal slum-, low-, middle- and high-income households without piped connection (73, 78, 94 and 112 L/capita/day) and with connection (91, 97, 113 and 133 L/capita/day), respectively. The collected data have been used to develop 20 statistical models using the multiple linear stepwise regression method for selecting the best predictor variable from the data set. It can be seen from the values that the strongest significant relationships of per capita consumption are with the number of occupants (R = −0.728) in the household and time spent to fetch water for use (R = −0.711). Furthermore, the results reveal that the highest fraction of end use is showering (18%), then bathing (16%), followed by toilet use (14%). This is not in agreement with many developing countries where toilet use represents the largest component of indoor end use.

**Keywords:** per capita water consumption; seasonal variation; water end-uses; Freetown; stepwise regression

## 1. Introduction

Seasonal water supply for domestic consumption is a significant concern that affects water demand in households [1]. Ensuring a sustainable water supply to the world's population between now and 2030 is a major challenge because of the current depletion of water resources and increasing global population size. In low-middle income cities, with the unreliable access and availability of piped water infrastructure, many households have resorted to alternative sources of improved water supplies. This can be linked to unpredictable weather patterns and the high demand on portable water resources [2]. It requires an understanding of the factors affecting water consumption and technical evidence of individual consumption which can be used to design an improved water demand projection [3].

Availability of freshwater sources is one of the biggest challenges facing the water sector in many developing cities, and this is exacerbated by the increasing material comfort of

the urban population per capita water demand that stresses the limited water resources [4]. Many studies [5–7] have analysed some of the factors impacting domestic water demand in developed and developing countries. The number of occupants, household type, household size, use of appliances, and presence of a swimming pool and evaporative cooler have been identified as variables that contribute to the variability of household water use in Melbourne, Australia [8]. In Makurdi, Nigeria [9], multiple regression analysis was used to identify seven variables, mainly household size, gender, number of children in household, kitchen type and level of education, as the significant factors influencing residential per capita water consumption.

Many factors affect per capita water consumption and these are mainly variations in rainfall patterns, effectiveness of the water sector, water quality, low pressure, residents' economic status, attitudes to water use, energy for water supply, water tariffs and management policies [4,10–13]. Consumer affordability is also influenced by the cost of the water supply [13].

Attaining water security for the world's population is a significant challenge especially in sub-Saharan Africa and Sierra Leone in particular. The rise in Freetown's population, intermittent water supply and inadequate dam infrastructure coupled with seasonal variability is visible from the long queues at water points, with varying sizes of collection containers and the long distance travelled mostly by women and children in search of water [10,14].

This paper compared dry and rainy season water sources available for consumption aimed at providing the first estimate of per capita water end-use volume in the different income groups in Freetown city, Sierra Leone. The study used a questionnaire survey to gather household data for a total of 398 households. The project includes using the collected data to develop statistical models using the stepwise regression method for selecting the best predictor variable.

## 2. Methods for Data Collection

Study Area: Freetown city, the national capital of Sierra Leone, covers an area of 81 km$^2$ (Figure 1). Data from the Sierra Leone Statistics Office indicated 229,951 households with a population of 1.055 million in the urban and sub-urban neighbourhoods [15]. The primary source of water supply in Freetown is piped water from the Guma Valley Water Company (GVWC), which is the only service provider. GVWC is a parastatal institution that is 99% owned by the Government of Sierra Leone and 1% by the Freetown City Council [16]. Alternative sources or multiple household water sources (MHWS) [17] include water stored in tanks, wells, boreholes, springs, gravity pipes, streams pond rivers, etc. In the rainy season, rainwater is collected in buckets under the roof or through structures that divert rainwater into large tanks and containers. Freetown experiences the African monsoonal rainfall type, which can be very torrential and makes it difficult to collect rainwater.

The water sector is facing serious challenges due to infrastructural dilapidation, settlement on catchment areas, increasing water demand from population growth, poor energy supply, and seasonal and climatic variability [7,18,19]. Despite the increasing population, the dam has not seen any improvement for its water storage capacity since its construction in 1960s. The provision of water supply is a significant challenge with piped water allocation to only 50% of Freetown's population [20].

The study area has been divided into four income groups based on their settlement areas as shown in Figure 1 [21]. There are deprived densely populated informal slum settlements; concentrated mainly along the coastal plains and marginalised land in the city (slums), these communities rarely have piped water to them, but have benefitted from other improved sources such as protected spring and gravity sources. Then, there are poor dense areas with less access to public standpipes. Thirdly, there are clusters of poor households in better-off areas with limited or no piped water supply services, either because of low pressure or absence of the system. The final group is the better-off neighbourhoods, with a meter piped connection [22].

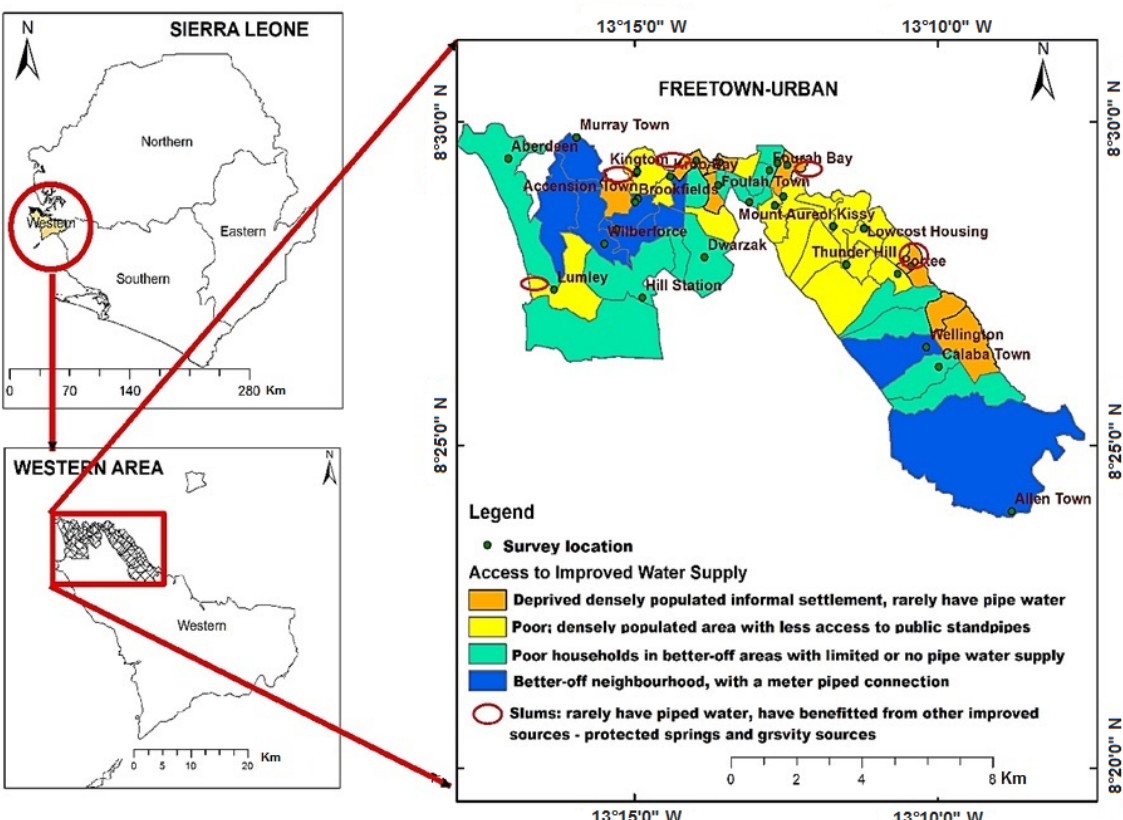

**Figure 1.** Map of study area with improved access to piped connection where research is conducted.

Data Collection Survey: Data for this research were collected using multiple-choice-format questionnaires containing over 80 questions. University students currently residing at their respective households were trained on how to measure flow rate, duration and water end-use volumes using a container and stopwatch to respond to the questions accurately on daily water end-use. The respondents were given two weeks to accurately record their water consumption use for any end-use activity conducted in the day recorded in litres. The total volume of water collected was also recorded. In adding up the total water end-uses, recycled water used was eliminated. The questionnaire was validated by 20 randomly selected households before the survey was conducted. A total of 550 questionnaires were distributed in August 2017 and April 2018 for the rainy (245) and dry (153) season survey, respectively. The full sample of both surveys consisted of a total of 398 households. The key variables investigated include the socio-demographic characteristics (age, gender, education and income); physical characteristics (number of rooms, vehicles, bathrooms, toilets and built-up area); water-use habits and ease of access (indoor volume, outdoor volume, collection containers, time to fetch, distance to source, water storage facility).

Data Analysis: In this survey, 398 surveyed questionnaires were received, coded and imported into IBM SPSS statistics V25.0 for analysis. The investigated households were categorised into four household income groups: Informal slum settlement, low-, middle and high-income households. The classification for the different income groups was based on the access to pipe-borne water and payments of water tariffs, defined by the Local Governance Act [23]. The water tariff for households in Freetown is determined from an evaluation of the house structure and location by the Freetown City Council and the Guma Valley Water Company assessment of the plumbing structure in the house based on the facilities included, e.g. shower, bathtubs, inside running taps or absence thereof. These income groups were evaluated individually to determine their daily per capita water consumption in litres per day (L/p/d). MS Excel was used to present the results in charts and table format. University students were identified to complete the questionnaires on

behalf of their households because they were in English. The investigated households were categorised into four household income groups and were analysed separately to determine their daily per capita water consumption in litres per day.

Development of statistical models: Using the dataset, 20 statistical models were developed using the multiple regression (stepwise) technique to select the best combination of household, socio-economic and water-use characteristics to construct the best-fit model based on strong statistical foundations.

## 3. Results and Discussion

### 3.1. Household Socio-Economic Characteristics

The analyses of household characteristics of 398 residential units revealed 60% of houses, 30% apartments and 10% of compound houses (rooms). The results show that 51% of the households (HHs) surveyed are middle-income, while the remaining 24%, 17% and 8% are low-income, high-income and informal slum settlements, respectively.

The survey revealed that only 33% of households have private connections to a piped water supply. The piped water supply is rationed on alternate days during both seasons throughout the study area and supplied for less than twenty-four hours. It is the primary source for households where it exists. Table 1 presents the households' percentage access to the different MHWS in the rainy and dry seasons. The table shows an increase or decrease in percentage use by households during the dry season in the study area. These sources include small-scale water sellers using pushcart to sell water in 22 litre jerry-can containers referred to in this research as vendor water, and water sold by tanker truck bowsers referred to here as tanker bowser water [17,24]. The pattern by which the households access their water sources showed that the middle- and high-income groups have the highest access to piped water and other water sources like bowser and bottled. Rainwater, gravity/spring and surface water are the prominent improved and unimproved sources for the lower-income households in the area, as well as for all households during the dry season when taps are closed for longer periods [25]. Water stored in tanks is provided and paid for communities' use by Nongovernmental Organisations (NGOs) and distributed to ten-thousand-litre containers stationed at certain deprived stand piped points in the study area. An example of this is shown in Figures 2 and 3. Packaged water is water sold in sachets, purported to be of better quality from water cottage industries mainly for drinking purpose. Some households reported saving rainwater in containers for use in the dry season, as well as households using packaged water for cooking light meals. At every improved and unimproved communal water point, households maintain a system to distribute equitable fetching and water collection time.

**Table 1.** Household percentage use of multiple water sources in the rainy and dry season for different water end-uses.

| Service Facility Types | Multiple Water Use Type | Rainy Season | | | | | Dry Season Increased or Decreased Change Access | | | | |
|---|---|---|---|---|---|---|---|---|---|---|---|
| | | Drink | Cook | Bathe /Hand Washing | Clothes Wash | Toilet Use, House Cleaning and Others | Drink | Cook | Bathe /Hand Washing | Clothes Wash | Toilet Use, House Cleaning and Others |
| Un-improved | Unprotected springs | 9% | 15% | 21% | 17% | 30% | +13% | +13% | +12% | +18% | +8% |
| | Unprotected dug wells | 20% | 5% | 13% | 13% | 28% | +3% | +8% | +10% | +15% | +14% |
| No service | Surface water (dam, streams, rivers, brook, pumping station) | 7% | 8% | 13% | 33% | 37% | +14% | +7% | +25% | +23% | +26% |
| Improved | Piped water | 45% | 89% | 88% | 86% | 82% | +26% | −3% | −6% | −8% | −12% |
| | Protected dug wells | 23% | 14% | 25% | 22% | 17% | +4% | +13% | +10% | +12% | +18% |
| | Boreholes | 16% | 20% | 22% | 24% | 33% | +3% | +13% | +16% | +16% | +16% |
| | Protected springs | 9% | 14% | 16% | 13% | 6% | +10% | +13% | +9% | +12% | +18% |
| | Rainwater | 38% | 93% | 96% | 95% | 96% | 0 | −2% | −3% | −3% | −3% |
| | Packaged water | 78% | 6% | 0 | 0 | 0 | +15% | +5% | - | - | - |
| | Bottled water | 25% | 0 | 0 | 0 | 0 | +5% | - | - | - | - |
| | Vendor water | 3% | 6% | 10% | 21% | 13% | +7% | +17% | +8% | +4% | +13% |
| | Tanker bowser | 2% | 42% | 55% | 32% | 45% | +11% | +6% | −2% | +6% | +6% |
| | Water stored in tanks | 15% | 32% | 38% | 34% | 40% | +9% | +2% | +6% | +2% | −5% |

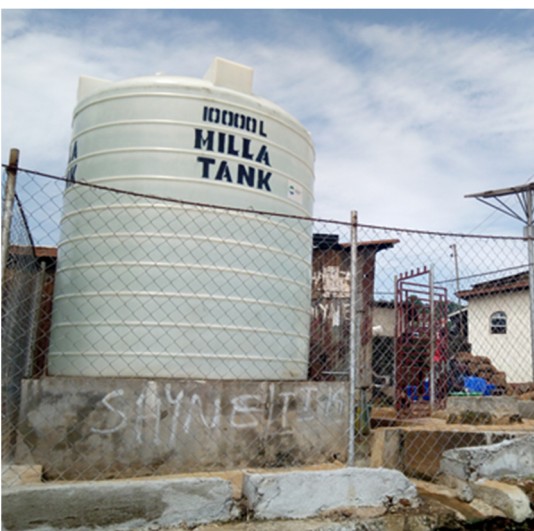

**Figure 2.** Water provided in 10,000 L tank at Allen Town.

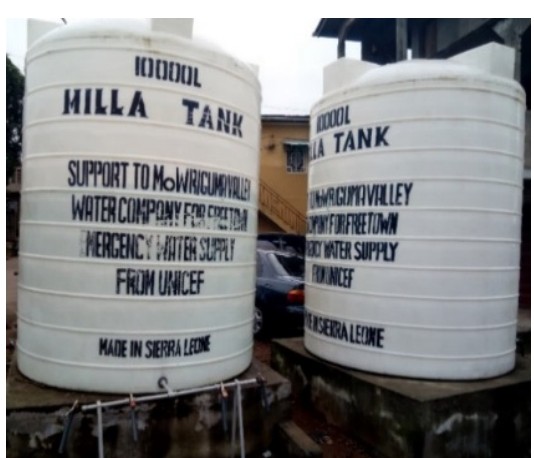

**Figure 3.** A 10,000 L tank located next to a public standpipe point at Congo Town.

A summary of the analyses of household and socio-economic characteristics of the 398 household units surveyed is shown in Table 2. It revealed that the average family size of all surveyed households was found to be 4.69 persons, approximately equivalent to the average household size (4.60 persons) by the Sierra Leone Population and Housing Census conducted in 2015 for Freetown. In this survey, the average number of adult males and adult females from 15–65 years were 1.35 and 2.06 per household, respectively. The average number of young (both male and female under 14 years), elders (65–75 years) and the aged (>76 years) were 0.97, 0.21 and 0.13 per household, respectively, showing great variation between the young and the old population [26].

The overall socio-economic characteristics of the surveyed households indicated an average floor area of 311 $m^2$ with a garden space of 32 $m^2$ for most of the surveyed households. In the surveyed households, 53% was a single storey, 30% were 2-storey, 8% were 3-storey, 5% were 4-story and 4% were 5-storey. The average number of rooms was three. The variation in the household family income was significant and ranged from $9 \times 10^5$ Sierra Leonean Leones (SLL)/month (≈£85) to Le $17 \times 10^6$ SLL/month (≈£1600), with an average household income equivalent to $5 \times 10^6$ SLL/month (≈£442). The monthly average family income is broadly consistent with the Government of Sierra Leone Civil Service Code: Regulations and Rules governing income and salary scales and the UN Salary scales for staff in the General Service and related categories [27].

**Table 2.** Summary of statistical parameters of household characteristics for the whole survey (398 households).

| Household Characteristics | Unit | Mean (Variance) | Sierra Leone Statistics Survey (2015) |
|---|---|---|---|
| Household size (occupancy) | No./hh | 4.69 (2.51) | 4.60 |
| Number of children (<14 years) | No./hh | 0.97 (0.84) | 0.90 |
| Number of adult male members (15–65 years) | No./hh | 1.35 (0.83) | 1.21 |
| Number of adult female members (15–65 years) | No./hh | 2.06 (1.14) | 2.00 |
| Number of elders (66–75 years) | No./hh | 0.21 (0.18) | 0.32 |
| Number of elders (>76 years) | No./hh | 0.13 (0.11) | 0.20 |
| Number of rooms in the household | No./hh | 3.31 (1.44) | 3.00 |
| Number of floors in the household | No./hh | 1.17 (0.93) | 1.00 |
| Total built-up area of floors | $m^2$/hh | 311.36 (4377.1) | 280.00 |
| Garden area per household | | 32.03 (160.38) | 28.00 |
| Monthly per capita income | SLL/mon ($\times 10^6$) | 1.35 ($1.43 \times 10^6$) | 0.90 |
| Household type | % | Houses (60.6%) Apartment (29.9%) Compound houses—rooms (9.5%) | Houses (54.4%) Apartment (20.2%) Compound houses—rooms (9.9%) |
| No. of houses, apartments and compound houses | No. | Houses (241) Apartment (119) Compound houses—rooms (38) | - |

Note: hh = household, SLL = Sierra Leone Leones (1000 SLL = £0.081).

### 3.1.1. The Effect of Household Socio-Economic Characteristics on the Average Total Water Consumption

The correlation coefficient R can be used to evaluate the strength of the relationship between variables [28,29]. The analysis of the data suggests a strong relationship between household occupancy (i.e., the number of people in the household) and total water consumption (R = 0.64), whilst there is a negative relationship between household total per capita consumption and household size (R = −0.728). The study revealed that family income has a positive correlation (r = 0.70, $p < 0.05$). This relationship implies that there is an increase in per capita water consumption with the monthly income. Per capita water consumption increases with the number of containers (R = 0.61) used by households but is negatively affected by the distance to water points (R = −0.53) and the time spent to fetch water and return back (R = −0.71). This finding is consistent with those of [30] who found that collection time and distance to water points are constraints to water access, because poorer households use less water as they have fewer storage containers and transport assets. In this study, variables such as education level and employment status provide some indications of the socio-economic status of the households in each income group. Generally, according to [31], the adult literacy rate in Sierra Leone is 32%. The high proportion of households with tertiary holders (42%) is because the surveys targeted university students to respond on behalf of their households, who have the requisite knowledge to understand and give accurate answers on access to their water supply. The results on occupation revealed that 31% were in trading and business, 29% civil servants, 24% artisans/craftsmen, and 16% were engineers, technicians, and surgeons.

a.  Distance to the sources of water

Although multiple water sources are available to the households, the surveyed households have preferences of particular sources for specific water end use either because of

availability or ease of access. The analysis of the full sample revealed that approximately 33%, 16% and 7% of the households have access of a household pipe connection, protected dug well and a borehole, respectively, within their households in the study area. Table 3 presents the percentages of the household's distance to access the multiple water sources in their neighbourhoods. Only 46% of the households obtained their water from a distance of 0–100 m to their homes. A total of 90% of the surveyed households fall within the UN stipulated distance of within 1000 m to their homes [32,33] while the remaining 10% of the households cover more than 1000 m in search of their water source. In such a situation, productive time is lost to trekking and queuing for long hours.

**Table 3.** Percentages of households with multiple water sources at various distances.

| Distance of Water Source to Homes (m) | HP$_C$ | BH | WS$_T$ | PD$_W$ | WB | VS$_S$ | PS | R/S | S/G | RW | Total Households (%) |
|---|---|---|---|---|---|---|---|---|---|---|---|
| 0–100 | 33 | 22 | 39 | 25 | 40 | 45 | 63 | 28 | 20 | 96 | 46 |
| 101–500 | - | 18 | 54 | 35 | - | - | 68 | 16 | 32 | - | 26 |
| 501–1000 | - | 10 | 39 | 23 | - | - | 50 | 14 | 13 | - | 18 |
| >1000 | - | 13 | 16 | 15 | - | - | 23 | 12 | 9 | - | 10 |

Note: PS—Public standpipe, WS$_T$—Water stored in tank, BH—Borehole, PD$_W$—Protected dug well, WB—Bowser, HP$_C$—household piped connection, RW—Rainwater, VS$_S$—Vendor pushcart, R/S—River/Stream, S/G—gravity/spring.

b. Time spent to water sources and return home

Figure 4 presents the percentages of households' distance and time spent to access their daily water use. For households a longer distance away from a water source, this affects the quantity of water collected for household use.

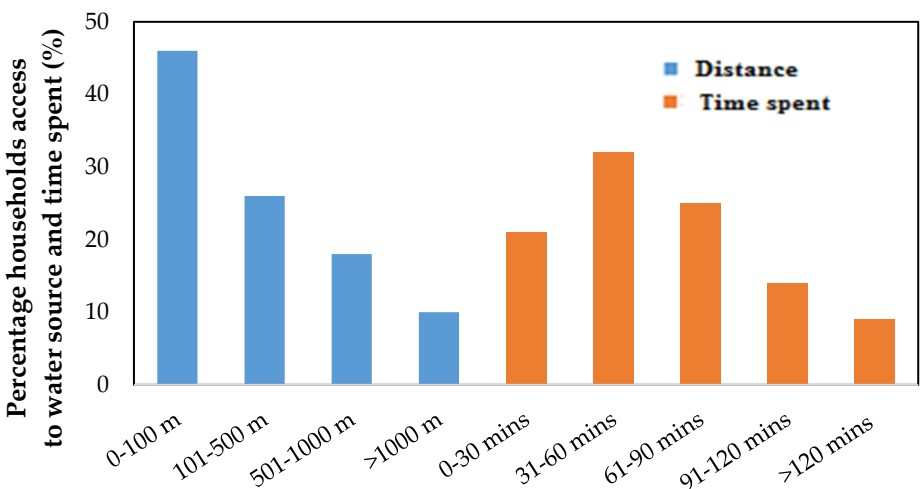

**Figure 4.** Percentages of households' distance and time spent to access their daily water use.

From the figure, only 21%, which is less than a third of the surveyed households, spend 30 min or less to access their water supply. The remaining 79% of households fall beyond the UN's recommended baseline time, which should not exceed 30 min to fetch water and return home [34]. The analysis revealed that productive time is lost to trekking and queuing for long hours to collect daily water use.

3.1.2. The Effect of Household Socio-Economic Characteristics on per Capita Average
Water Consumption

The frequency distribution of the total average daily per capita water consumption
for the whole sample is shown in Figure 5, signifying that the average is about 93 litres per
capita per day (L/p/d). The average daily per capita water consumption for households
with a piped connection is 112 litres per capita per day (L/p/d). These amounts are higher
than the nationwide estimated per capita volume by the [35] set at 40 L/p/d with only
piped water use. This total average water consumption is the volume of water obtained via
the various multiple sources, as indicated in Table 1. Per capita consumption varies from
73 to 112 (L/p/d) for households without pipe-water connections where showering, toilet
flushing and hand wash basin use are absent, and from 91 to 133 (L/p/d) where showering,
toilet flushing and hand wash basin tap use are common. The increase in male members
and children in the household increases per capita consumption. This increase in per capita
consumption for males seems to be because a high percentage of men are engaged in daily
employment and use water for personal hygiene, probably washing clothes more than
other members of the family daily. The high consumption for children is because they need
to be cleaned more often than adults or elders in the household.

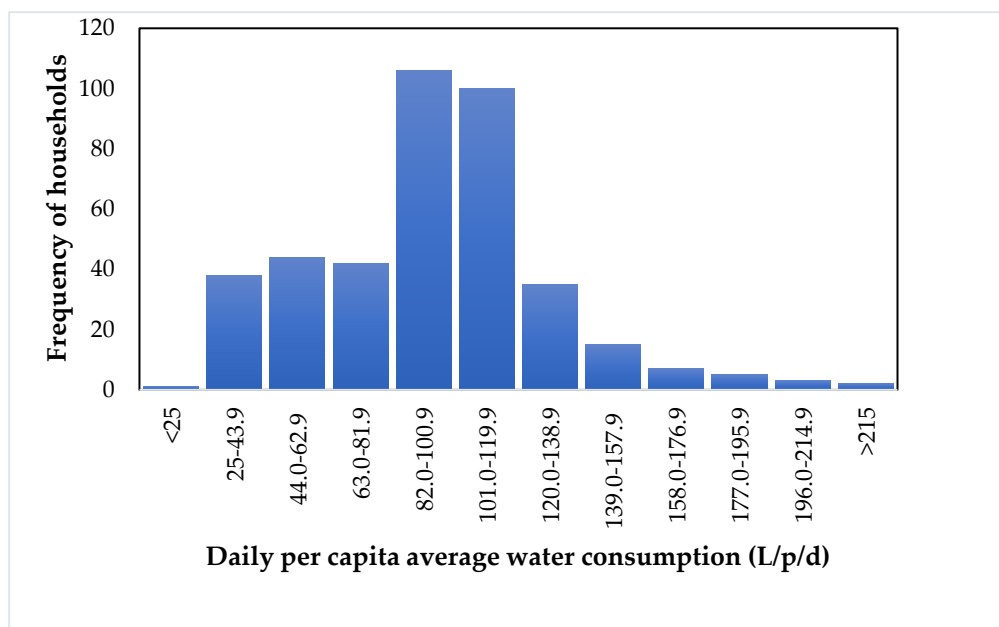

**Figure 5.** Frequency distribution of average per capita water consumption.

3.1.3. The Effect of per Capita Income on the Average Water Consumption

The results of each analysed group with either piped connection or nonpiped con-
nection reveal that the average daily per capita water consumption increases with income
levels (i.e., 73, 78, 94 and 112 L/p/d in informal settlements, low-, middle- and high-income
groups, respectively, for nonpiped households), with an average per capita water for the
full sample of 93 L/p/d. Households with some piped connections have indicated the
use of showers, wash hand basins and cistern toilets. The average daily per capita water
consumption also increases with income levels (i.e., 91, 97, 113 and 133 L/p/d in informal
settlements, low-, middle- and high-income groups, respectively). The distribution of water
use reveals slight variations between income groups (Figure 6a,b). Figure 6a shows that
the highest distribution is showering (21%), when it is done by occupants, then followed
by toilet flushing (16%) and clothes washing (15%). In Figure 6b, the highest distribution
fraction is bathing (22%), followed by laundry (18%) and toilet use (15%). These are in
contrast to many developing countries where toilet use consistently represents the largest
component of indoor end use [35].

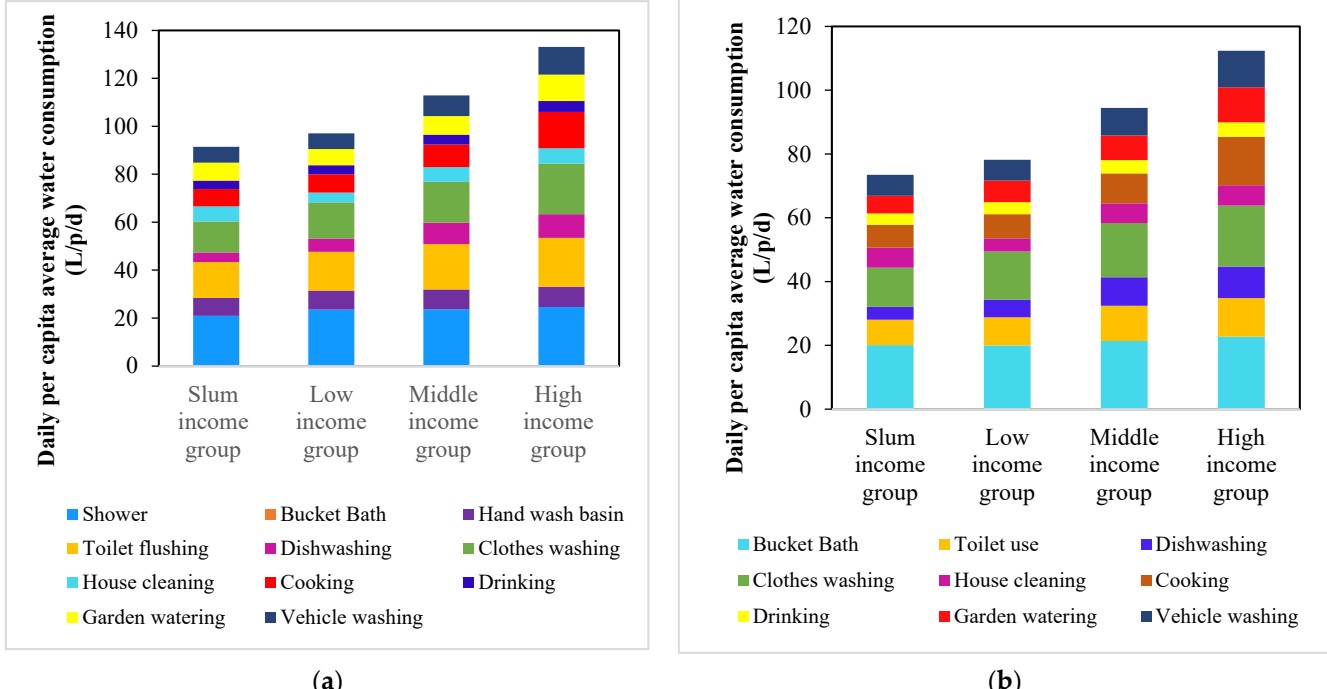

**Figure 6.** (**a**) Impact of per capita monthly income on water end uses in Freetown for households with piped-connection. (**b**) Impact of per capita monthly income on water end uses in Freetown for households without piped connection.

### 3.2. Average per Capita Water Use for the Different Water End-Uses (Micro-Components)

Here, a household's total water consumption is divided into a number of micro-components: Showering, bucket bath, toilet flushing, house washing, cooking, dish washing, clothes washing, wash hand basin, garden watering and vehicle washing. The distribution of average daily use of each of these components in all income groups is shown in Figure 6a,b. Only some of the households recorded shower, hand wash basin and cistern flush use. Of the 398 surveyed households, none were recorded to have a swimming pool. However, some households recorded owning a garden area (34%) and vehicle (61%). In agreement with [36], daily per capita consumption decreases with the number of household occupants.

The summary of average values for different micro-components per person (e.g. frequency, duration of use and flow rate) is illustrated in Table 4. It shows the comparison between these parameters in informal settlements, low-, middle- and high-income households. The water use characteristics for different micro-components of the households in different income groups are briefly discussed in the following sections below (Section 3.2.1 to Section 3.2.12).

**Table 4.** Summary of mean values of water end-use parameters (398 households).

| End-Use | Parameter/Variable | Unit | Overall Sample | Slum Income | Low Income | Middle Income | High Income | Comparison with Past Studies |
|---|---|---|---|---|---|---|---|---|
| Shower | Number of showers taken per capita per day | shw/p/d | 0.52 | 0.34 | 0.35 | 0.51 | 0.92 | (0.51 shw/p/d) [11] |
| | Duration of each shower | min/shw | 3.28 | 3.02 | 3.43 | 3.32 | 3.35 | |
| | Flow rate | L/min | 6.93 | 4.60 | 4.12 | 7.86 | 9.54 | (0.13–0.17 min/shw) [37] |
| Bathing (Bucket) | Number of baths taken per capita per day | bt/p/d | 0.94 | 0.92 | 0.92 | 0.94 | 0.96 | (0–1 bt/p/d) [38] |
| | Volume of water used in each bath | L/bt | 20.80 | 20.04 | 20.30 | 21.38 | 22.73 | (20L) [12] |
| Hand wash basins | Number of times hand wash basins are used per capita per day | brt/p/d | 3.07 | 3.20 | 3.02 | 3.00 | 2.72 | (10 brt/p/d) [11] |
| | Duration of tap use | sec/brt use | 59.55 | 58.17 | 57.41 | 59.29 | 62.00 | (3–4 brt use) [39] |
| | Flow rate | L/min | 2.65 | 2.41 | 2.73 | 2.80 | 3.02 | (2–5 L/min) [40] |
| Toilet flushing | Number of toilet flushes used per capita per day | tf/p/d | 3.05 | 3.19 | 3.25 | 3.16 | 2.61 | (2–7 tf/p/d) [41] |
| | Volume of water used per person in each toilet flush | L/tf | 5.71 | 4.37 | 4.33 | 5.81 | 7.02 | (4–10 L) [42] |
| | Number of latrines used per capita per day | lat/p/d | 2.96 | 3.11 | 3.04 | 3.04 | 2.80 | (1–3 lat/p/d) [42] |
| | Volume used per person for each pit use | L/lat/fl | 1.82 | 1.78 | 1.82 | 1.84 | 2.01 | (2–3 L) [30] |
| | Number of pour flush latrines used per capita per day | pf/p/d | 3.02 | 3.12 | 3.11 | 3.06 | 2.96 | |
| | Volume used per person for each pour flush use | L/pf/d | 2.8 | 2.77 | 2.83 | 2.95 | 2.90 | (2–3 L) [43] |
| Dishwashing (bowl) | Number of dish-washing per day | dws/d | 2.00 | 2.00 | 2.00 | 2.00 | 2.00 | |
| | Volume of water used in each dishwashing | vol/wsh | 6.91 | 6.02 | 6.52 | 8.57 | 9.99 | (15–23 L) [44] |
| House cleaning | Number of house-cleaning per day | wsh/d | 0.16 | 0.18 | 0.15 | 0.16 | 0.18 | (3–78 L) [45] |
| | Total volume used per household per day | L/p/d | 8.42 | 9.84 | 7.58 | 7.20 | 6.37 | (1–20 L) [30] |
| Clothes washing (hand) | Number of clothes-washing sessions | wsh/d | 0.26 | 0.28 | 0.29 | 0.26 | 0.21 | |
| | Volume of water used per wash per day | L/wsh/d | 16.50 | 17.85 | 18.64 | 19.51 | 21.10 | (5–20 L) [46] |
| Vehicle washing | Number of vehicles washed per day | wsh/d | 2.23 | 1.33 | 1.38 | 2.68 | 1.66 | |
| | Volume used per day | L/wsh/d | 10.38 | 9.60 | 9.51 | 10.77 | 10.78 | |
| Cooking | Volume of water consumed in cooking | L/p/d | 12.11 | 9.86 | 10.08 | 11.98 | 16.48 | (10–50 L) [38] |
| Drinking | Volume of water consumed for drinking | L/p/d | 4.17 | 3.56 | 3.88 | 4.18 | 4.51 | (2.7–3.7 L) [47] |
| Garden | Volume of water consumed for garden | L/p/d | 9.24 | 7.50 | 7.80 | 8.07 | 10.54 | (0.4 t/d) [48] |
| | Total water consumption | L/p/d | 93 | 91 | 97 | 113 | 133 | (28–244) for 5–50% piped households with MWSU access [25] |

Note: L/p/d = litre per person per day, L = litre, p =person, d = day, wsh = washes, min = minute, vol = volume, bt = bath, shw = shower, sec = second, brt = bathroom tap, tf = toilet flushing, lat = latrine, pf = pour flush, fl = flush, dws = dishwash, t = time, No./d = number per day, unit for all volume = litres.

### 3.2.1. Showering

Showering is only common to some households (47%) and has a positive relationship to household income. The daily per capita water use for showering is a function of the number of times taken, the duration and flow rate of shower. The number of times a shower is taken rises across income groups. The average number of showers in the full sample is moderate (0.52 shw/p/d), with an average flow rate of shower (6.93 L/min) (Table 4). Most of the households with piped connections recorded a low tap pressure flow of their water supply, especially in peak hours of the day. The specific shower types in the households were not investigated. The average duration is 3.28 min/shower, and the times a shower is taken increases with per capita income. Showering accounts for the highest (18%) distribution of indoor daily water end-use, but showers are only taken by households when tap water is available, as piped water is rationed throughout the study area.

### 3.2.2. Bathing (Bucket)

In all income groups, having a bucket bath is common to all households and accounts for 16% of total water use (Figure 6b). The results show a frequency of 0.92 to 0.96 per capita per day (Table 4). The average daily per capita use varies from 20 litres (L) in the lower-income groups to 23 litres (L) in the high-income group. The use of bathtubs is not a common practice in Freetown, because of the volume of water it will consume if members of households prefer to take baths in them. Generally, in all income groups, as the size of the household increases, the amount of water used for bathing per person decreases. The smallest household size (2 persons) has the highest water consumption per capita with the larger-size (8–12 persons) households having the lowest per capita usage [49]. The quantity of water required to maintain good hygiene may vary significantly depending on the water collection behaviour [29,36].

### 3.2.3. Toilet Use and Flushing

Based on the survey of the households involved in this study, the toilets were either pit latrines (52%), single flush with cistern (34%) or pour flush (14%) with average capacities of 1.8, 5.7 and 2.8 L respectively (Table 4). Toilet flushing refers to the use of these various toilet types. The calculated average toilet flush per capita per day was 2.8 times/day. The frequency of per capita toilet use was higher in the informal settlement slum (3.19 times/day) and low-income households (3.25 times/day) than in the other high-income households. The average number of occupants in the informal settlement group was 6.2. In [9], it was explained that the higher frequencies and volumes used by the lower-income level groups may be because of the squalid conditions in which these households live in. Therefore, they are at high risk of water-related diseases and would spend most of their time using the toilets. The low frequency in the high-income-level households may be because of low household size or that they spend most of their time during the day at the workplace, where some flushing at home is replaced by flushing at the workplace. From the data presented in Table 4, it appears that in the high-income households, water consumed for personal hygiene-related activities is still high because of their awareness to maintain healthy hygiene.

### 3.2.4. Hand Wash Basin Tap Use

The tap use considered in this study is water used in hand wash basin taps (teeth cleaning, hand washing, ablution, kitchen sink) where applicable. In all income groups, hand wash basin users are low, accounting for only 5% of total water use (Figure 6). Similarly to shower use, hand wash basin usage is influenced by the number of times the hand wash basin tap is used, and this is subject to when pipe water is available to the household. As with showers, the flow rates from the hand wash basin increases with household income. The reason for this could be that households in the higher-income group have better fitted plumbing structures to increase the flow to their homes. The frequency of hand wash basin use also rises with income. The average duration of hand

wash basin tap use for all income groups is 59 s per use. When multiplied with the number of times of hand wash basin tap use, the total daily per capita tap duration becomes 3.10, 2.89, 2.96 and 2.81 min/p/d for informal settlement, low-, middle- and high-income households, respectively. These figures are similar to values found in the literature of Victoria, Australia [39]. The analysis also showed that households with taps use more water per capita than those without [50].

### 3.2.5. Dishwashing

The use of a dishwasher is not common in Freetown, mainly because of the lack of energy and irregular water supply to operate it. None of the respondents recorded owning a dishwasher in both the rainy and dry seasons' surveys. Dish washing is mainly done manually in a bowl of water and mostly done at the household level. Per capita dishwashing accounts for 5% of the average total water usage. The daily water consumption for dishwashing is a function of the number of dish-washing a day and the volume of water used in each wash. The frequency of dish-washing is 0.51 per person per day for all income levels, i.e., after each meal (breakfast and dinner). There is a considerable mean difference in total per capita water use between households in the lower-income levels (6.02 and 6.52 L/p/d) for informal and low-income groups, respectively, and those in the higher-income groups as they use 8.57 and 9.99 L/p/d for middle- and high-income households, respectively (Table 4). Families in the lower-income groups are larger in number and they undertake certain activities (e.g. eating and sleeping) communally. Therefore, they may use less dishes and water than families in higher-income households.

### 3.2.6. Clothes Washing

Water-saving household appliances such as washing machines are not common in the survey area. The reason could be mainly because of the lack of energy to power the appliance and continuous water availability for its operation. Hence, clothes and dishes are done mostly by hand in a bowl of water as it is more efficient and inexpensive in this region [30]. Of the 398-sample survey, only two respondents in the middle-income level group recorded owning a washing machine but hardly use it because of a lack of constant energy and piped water supplies. Washing clothes by a washing machine can use from 40 to 200 L per wash depending on the technology [44]. It has been observed that washing clothes by hand in a bowl with water uses much less volume (20 L) and is more sustainable. The main parameters to identify water consumption for clothes washing are the number of times clothes washing is done per day and the volume of water used per wash. Clothes' washing is done from 0.21 times/day for the high-income group to 0.28, 0.29 and 0.26 times/day in informal settlement, low- and middle-income groups, respectively. Previous studies have observed that people with more clothes might not have to wash clothes more often as people with fewer clothes [45].

Other parameters that can influence the number of clothes washing per household per week can be seasonal (temperature) variability and the number of occupants in the household [51]. Clothes washing can become more frequent in hot and dusty weather [52]. The average per capita water use is 18, 19, 20 and 21 L/p/d in informal settlement, low-, middle- and high-income families, respectively.

### 3.2.7. House Washing

Analysis of daily average water use for house cleaning is shown in Table 4 and Figure 6. It can be seen that a slight variation exists in daily volume used among the households. The volume for house washing constitutes about 5% of the total daily water consumption. The average quantity of 8 L/p/d could be because of the many multitenant apartment and compound houses (rooms) present in the area. These multitenant households usually share communal space, e.g. toilets, kitchen spaces, if present, and room sizes are usually small, and so do not require much water for cleaning activities. Cleaning activity is mostly done with water in a container. The frequency of cleaning is from 0.15 to 0.18 times/day

(Table 4). Most of the high-income households have their floors carpeted or covered in linoleum mats, which uses less water to clean.

### 3.2.8. Cooking

The per capita water consumption per day in developing countries can be as low as 20 L [53]. The UN also noted that a human being needs 50 L of water per day in order to prepare meals and to have enough for personal hygiene. The current study shows that the average volume of water required to prepare food increases with family income, accounting for 9.86, 10.1, 11.9 and 16.5 L/p/d in informal settlement, low-, middle- and high-income households, respectively (Table 4).

### 3.2.9. Drinking

It is evident that between 1990 and 2008, about 1052 million of the urban population in developing countries have gained access to improved drinking water despite struggling with the population growth to have equitable distribution [54]. However, there are still challenges for water managers because the number of urban dwellers living in slum-deprived areas continues to increase with limited service provision [15,48]. By 2025, half of the world's population will be living in water-stressed areas [55]. In this survey, drinking accounts for 3% of the total household water consumption in this survey. The average per capita drinking consumption is 4 L per day, which is slightly above the 2.7–3.7 L designated by [56]. The analysis revealed that 37% of the respondents are concerned with the quality of the water they collect and 48% of all the households explained that they perform some form of treatment such as adding sterilising tablets or leaving it to settle in a special container before use. Therefore, more than 75% of the total households explained that they prefer to consume packaged water, that is, water in plastic sachets or bottled which they believe has been properly prepared for consumption. This is in line with the [57] observation who explained that more than 35% of households consume bottled and package water at home in consideration for quality.

### 3.2.10. Outdoor Water Usage

The outdoor use is composed of garden watering, swimming pool usage and vehicle washing. No information on the outdoor activity swimming was recorded in the whole sample. This could be due to several reasons such as a decrease in temperature condition, and economic and physical water scarcity.

### 3.2.11. Vehicle Washing

The analyses show that in the case of vehicle washing, the highest frequency of vehicles washed per day is the middle-income group (2.68 wsh/d). However, in terms of volume of water used per wash per day, the highest consumers are the high-income households. The consumption of average daily per capita water use of 10.38/wsh/d is because of seasonality and availability of water sources for use. It can be seen from the data in Table 4 and Figure 6a,b that the average per capita water use for vehicle washing accounts for 7% of the total daily water consumption. Water used for vehicle washing is collected from the MHWS (viz. tap, rain, wells, streams, tanks and springs). None of the households recorded using a water hose for washing vehicles. Some households also indicated that their vehicles are sometimes washed at car washing centres.

### 3.2.12. Garden Watering

In terms of garden watering, and like most of the other end-uses, none of the households recorded using a water hose. Most of the houses recorded only one watering session per day, either in the morning or in the evening. During the rainy season survey, none of the households recorded water consumption for gardens. This may be because they depend on the rain to water their gardens. In order to measure the seasonality impact, the

survey was repeated during April (2018) to account for water consumption variations in the dry season.

The total volume of water used for garden watering increases slightly with income levels: 7.50, 7.80, 8.07 and 10.54 L/p/d in the informal slum-, low-, middle- and high-income households, respectively.

*3.3. Statistical Modelling of Daily per Capita Water Usage with Household Socio-Economic Characteristics*

The water consumption data from the full 398 households were divided into calibration and validation sets. Then, 70% of the data were used for calibration (i.e., training), while the remaining 30% were spared for validation (i.e., testing) purposes. The calibration data set was used to develop statistical models to predict per capita consumption as a function of household socio-economic characteristics. The household socio-economic characteristics were divided into three groups, that is:

1.  Socio-demographic characteristics: e.g., number of children, adult females, adult males, elders 66–75 years and elders over 76 years.
2.  Physical characteristics: e.g., the number of rooms, household size, the total area of floors and house type.
3.  Water-use characteristics: e.g., shower volume, toilet flushing volume, time spent to fetch water and distance to water source.

Models Based on Multiple Linear Regression (Stepwise)

The STEPWISE multiple regression approach has been previously used successfully to predict water demand [11]. The technique readily selects the combination of relevant independent variables to develop the best-fit model based on strong statistical foundations and saves on the intense computational effort required by some other methods (e.g. evolutionary polynomial regression). It is a potential approach for selecting the best predictor variable from a large number of variables.

The Stepwise multiple regression approach is applied using IBM SPSS Statistics (v. 25) software to determine the best subset model for daily per capita water use estimation. Using the calibration set of data, the relationship between the independent variables (household socio-economic and water use characteristics) and the dependent variable (per capita water consumption) was investigated, and the values of the correlation coefficient (R) are shown in Table 5. From the table, it can be seen that the strongest significant relationships of per capita consumption are with the number of occupants (R = −0.728) in the household and time spent to fetch water for use (R = −0.711).

The acceptance or deletion of an independent variable for the regression model is based on the strength of the relationship (i.e., the strength of the correlation) and also its contribution to the decrease in the residual sum of squares [11]. The regression coefficients and model are then statistically verified at every iteration to select or delete the independent variable.

Using the STEPWISE approach with the calibration set of data of the 398 investigated households, four models were developed based on demographic, physical, water use and whole characteristics (i.e., Model 1, 2, 3 and 4 in Table A1). The same process is repeated using the calibration set of informal slum-, low-, middle- and high-income households' data. These models are also shown in Table A1 and they are statistically significant at ($p < 0.05$).

In total, 20 models were developed. The predictions from these models were plotted against the actual per capita water consumption values obtained from the study, as shown in Figure A1. The figure shows that the trend-lines of validation and calibration sets are almost indistinguishable in all cases. Additionally, the $R^2$ value improves further when the water consumption data are disaggregated into the various income groups, i.e., informal slum, low, middle and high.

**Table 5.** Correlation coefficients between household characteristics and per capita water consumption.

| | | Correlation Coefficient Value (R) | | | | | | | | | | | | |
| | | Demographic Characteristics | | | | | Physical Characteristics | | | | Water Use Characteristics | | | |
| | | No. of Children | No. of Adult Females | No. of Adult Males | No. of Elders 66–75 ($E_{66-75}$) | No. of Elders >76 ($E_{>76}$) | No. of Occupants | No. of Rooms | No. of Floors | Total Built-Up Area ($m^2$) | Shower Volume (L) | Toilet Flush Volume (L) | Distance to Source (m) | Time Spent (min) |
|---|---|---|---|---|---|---|---|---|---|---|---|---|---|---|
| **Per capita water consumption (L/p/d)** | All investigated households | −0.527 | −0.593 | −0.512 | −0.534 | −0.251 | −0.728 | −0.163 | −0.056 | 0.021 | 0.631 | 0.562 | −0.531 | −0.711 |
| | Informal slum households | −0.605 | −0.721 | −0.534 | −0.527 | −0.283 | −0.760 | −0.204 | −0.501 | 0.319 | 0.582 | 0.673 | −0.745 | −0.763 |
| | Low-income households | −0.590 | −0.654 | −0.403 | −0.364 | −0.273 | −0.758 | −0.261 | −0.426 | −0.427 | 0.675 | 0.635 | 0.782 | −0.731 |
| | Middle-income households | −0.648 | −0.683 | −0.493 | −0.379 | −0.261 | −0.783 | −0.593 | −0.413 | −0.526 | 0.719 | 0.630 | −0.664 | −0.726 |
| | High-income households | −0.572 | −0.650 | −0.484 | −0.243 | −0.252 | −0.819 | −0.673 | −0.529 | 0.537 | 0.720 | 0.743 | −0.745 | −0.718 |

Note: L/p/d = litres per capita per day.

## 4. Seasonal Variability and Impact of Average per Capita Water Consumption

### 4.1. *Average per Capita Water Consumption in Rain Season*

The seasonal variability of domestic water consumption in many tropical countries is mainly affected by climate, seasonal and hydrological conditions [57–61]. Per capita water consumption is a function of socio-economic, weather, season, hydrological characterisation, lifestyle and technical factors. It varies with people's behaviour, habits, income level and culture. Therefore, per capita water use varies from one region to another [17,19].

Daily per capita water consumption was found to be about 7% higher than the average daily consumption for the full sample in the rainy season, whilst daily per capita water consumption was almost 14% lower than the full survey in the dry season.

The frequency distribution and cumulative frequency of per capita average water consumption for all surveyed households during the rainy and dry season are shown in Figure 7. From this figure, it can be seen that the number of households which consume more than 93 L/p/d is decreased from 71% in the rainy to 6% of households in the dry season. Further analysis of the dry season survey shows that the daily per capita average water consumption is mainly between 26 and 75 L/p/d compared to that in the rainy season, which is between 75 and 120 L/p/d. Additional analysis revealed that the majority of the consumption is lower in the dry season because of the water scarcity and limited access to alternative water sources. The analysis revealed that productive time is lost to trekking and queuing for long hours to collect daily water for use. These values of both seasonal surveys are not in agreement with those of the [32] report, which showed that per capita consumption ranges between 40 and 78 L/p/d during both seasons in the year.

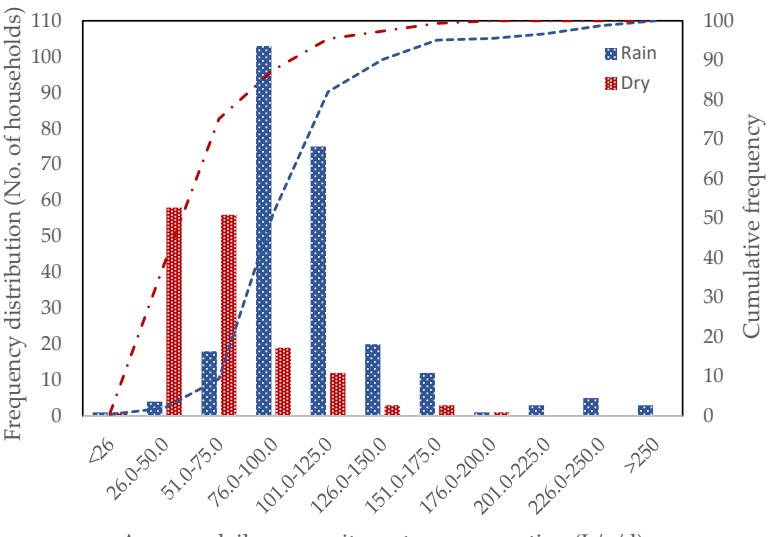

**Figure 7.** Seasonal variability of per capita average water consumption.

### 4.2. *Seasonal Variability of Water End-Use*

Households access their domestic water for consumption based on their income levels, accessibility and consumption patterns [62] The piped connected households receive water for a short period of time on some days of the week. Hence, households have responded by storing water in large containers in the home, as well as using all available MHWS, which can be improved private wells/boreholes and other unimproved sources (unprotected well, unprotected springs, stream) [63].

To study the seasonal variability of water end-uses, a two-tailed t-test is used at a 95% confidence interval, as shown in Table 6. It can be observed from this table that the *p* value of bathing, cistern flushing, latrine use, hand wash basin taps, pour flush use and house cleaning is higher than 0.05. This explains that there is no statistically significant difference between the consumption in the rainy and dry season. These finding are in agreement

with [38,58,59], which showed that toilet use and bathing are less sensitive for seasonality. Conversely, the other water end-uses (i.e., shower, dishwashing, clothes washing, drinking, cooking, vehicle washing and garden watering) have a statistically significant difference ($p < 0.05$) between the two seasons (Table 6).

**Table 6.** Statistical comparison of water end-uses between rainy and dry season.

| Water End-Use | Average Water Consumption (L/p/d) | | Mean Different (Rain-Dry Season) | Percentage Difference | t Value | Significant (2-Tailed) ($p$) |
|---|---|---|---|---|---|---|
| | Rainy | Dry | | | | |
| Showering | 30.00 | 21.83 | 8.17 | 53.2 | 2.243 | 0.026 |
| Bathing | 20.70 | 16.50 | 4.2 | 27.3 | −1.062 | 0.290 |
| Hand wash basin | 8.04 | 5.19 | 2.85 | 18.5 | −1.043 | 0.299 |
| Cistern flushing | 14.37 | 11.37 | 3.0 | 19.5 | 1.009 | 0.315 |
| Latrine use | 5.68 | 5.40 | 0.28 | 1.8 | −1.705 | 0.090 |
| Pour flush use | 8.96 | 8.41 | 0.55 | 3.6 | 1.232 | 0.220 |
| Dishwashing | 8.40 | 7.64 | 0.76 | 4.9 | 8.514 | 0.000 |
| Clothes washing | 19.25 | 15.43 | 3.82 | 24.9 | 2.827 | 0.005 |
| Drinking | 4.38 | 3.78 | 0.6 | 3.9 | −2.244 | 0.026 |
| Cooking | 10.83 | 14.15 | −3.32 | −21.6 | 4.121 | 0.000 |
| House cleaning | 8.96 | 6.60 | 2.36 | 15.4 | −0.150 | 0.881 |
| Vehicle washing | 11.53 | 10.25 | 1.28 | 8.3 | 1.276 | 0.020 |
| Garden watering | 0.00 | 9.18 | −9.18 | −59.7 | −2.695 | 0.013 |

Note: $p < 0.05$ = significant difference between rainy and dry. $p > 0.05$ = no significant difference between rainy and dry.

An efficient technique of studying and estimating per capita water consumption is to separate the various water end-uses into component parts [37]. During the dry season, indoor water use (105.4 L/p/d) decreases compared to rainy season consumption (115.8 L/p/d) for households with piped connections (Table 6), whereas outdoor use (vehicle washing and garden watering) shows a slight seasonal increase from 11.53 L/p/d in the rainy season to 19.43 L/p/d in the dry season. Similarly, in the dry season, for households without a piped connection and with either a latrine or pour flush toilet, indoor water use (72.4 and 75.6 L/p/d) decreases compared to rainy season consumption (89.7 and 93.0 L/p/d) for latrine and pour flush toilet types, respectively. The outdoor use (vehicle washing and garden watering) maintains the same seasonal increase.

The summary of average values of water end-use parameters (number of uses, duration of use, flow rate where applicable and volume) is illustrated in Table 7. The table shows the comparison of these parameters between the rainy and dry season.

*4.3. Seasonal Variability Impact on Water End-Uses between the Seasons*

The reliability of piped water supplies has deteriorated in the dry season. Showering is only common to some households with piped water connection (44%) and increases with family income. Showers were the greatest end-uses among households, accounting for 16–20% on average for the dry and rainy season, respectively, and ranging between 13% and 22% in the lower- and upper-income levels.

The comparison of rainy and dry surveys showed that 100% of the households in the dry season consume between 18 to 25 L/p/d for showering, whereas in the rainy season, only 28% of households consume 25 L/p/d or less, with the remaining 72% consuming above 25 L/p/d. There is a reduction in the number of households during the rainy (15%) and dry (13%) season. The decrease in shower water use and number of households taking a shower is attributed to water scarcity. However, the average duration of each shower decreases from 3.61 min/shower in the rainy to 2.36 min/shower in the dry season, with an increase in shower flow rate of 7.02 L/min in the rainy to 9.25 L/min in the dry season (Table 7). This finding is consistent with the [64] explanation, which stated that household activity water usage can vary greatly depending on associated technology and water availability.

**Table 7.** Statistical variability of mean values of water end-uses parameters.

| End-Use | Parameter/Variable | Unit | Overall Survey | | Slum Income | | Low Income | | Middle Income | | High Income | |
|---|---|---|---|---|---|---|---|---|---|---|---|---|
| | | | Rainy | Dry | Rainy | Dry | Rainy | Dry | Rainy | Dry | Rainy | Dry |
| Shower | Number of showers taken per capita per day | shw/p/d | 0.39 | 0.70 | 0.36 | 0.36 | 0.33 | 0.44 | 0.40 | 0.76 | 0.51 | 0.94 |
| | Duration of each shower | min/shw | 3.61 | 2.36 | 3.40 | 2.50 | 3.30 | 2.55 | 3.80 | 2.36 | 4.38 | 2.37 |
| | Flow rate | L/min | 7.02 | 9.25 | 5.95 | 7.20 | 5.30 | 7.61 | 7.36 | 9.42 | 9.49 | 9.87 |
| Bathing (Bucket) | Number of baths taken per capita per day | bt/p/d | 0.95 | 0.97 | 0.94 | 0.95 | 0.91 | 0.96 | 0.95 | 0.98 | 0.94 | 0.99 |
| | Volume of water used in each bath | L/bt | 20.70 | 16.50 | 19.80 | 9.50 | 18.20 | 11.03 | 20.80 | 18.25 | 25.00 | 20.46 |
| Hand wash basins | Number of times hand wash basins are used per capita per day | brt/p/d | 3.06 | 2.06 | 3.36 | 2.30 | 3.02 | 2.20 | 3.29 | 2.22 | 3.57 | 1.90 |
| | Duration of tap use | sec/brt use | 60.62 | 57.09 | 58.00 | 56.00 | 57.00 | 57.00 | 58.93 | 58.07 | 62.00 | 57.00 |
| | Flow rate | L/min | 2.63 | 2.65 | 2.51 | 2.51 | 2.47 | 2.53 | 2.64 | 2.64 | 2.68 | 2.69 |
| Toilet flushing | Number of toilet flushes used per capita per day | tf/p/d | 3.11 | 2.52 | 3.13 | 3.00 | 3.07 | 2.84 | 3.04 | 2.51 | 3.23 | 1.80 |
| | Volume of water used per person in each toilet flush | L/tf | 4.80 | 4.51 | 4.30 | 4.15 | 4.25 | 4.25 | 4.80 | 4.60 | 5.20 | 5.00 |
| | Number of latrines used per capita per day | lat/p/d | 2.6 | 3.00 | 3.4 | 3.15 | 3.3 | 3.22 | 3.1 | 3.01 | 2.9 | 2.72 |
| | Volume used per person for each pit use | L/lat/fl | 1.8 | 1.80 | 1.7 | 1.80 | 1.8 | 1.95 | 1.9 | 1.81 | 2.1 | 2.01 |
| | Number of pour flush latrines used per capita per day | pf/p/d | 3.20 | 3.00 | 3.35 | 3.12 | 3.29 | 3.27 | 3.17 | 2.80 | 2.98 | 2.52 |
| | Volume used per person for each pour flush use | L/pf/d | 2.8 | 2.85 | 2.5 | 2.72 | 2.5 | 2.98 | 3.0 | 2.92 | 3.0 | 3.00 |
| Dishwashing (bowl) | Number of dish-washing per day | dws/d | 2.00 | 2.00 | 2.00 | 2.00 | 2.00 | 2.00 | 2.00 | 2.00 | 2.00 | 2.00 |
| | Volume of water used in each dishwashing | vol/wsh | 8.40 | 7.64 | 7.83 | 3.38 | 6.52 | 4.60 | 8.70 | 8.01 | 8.70 | 10.71 |
| House cleaning | Number of house cleaning per day | wsh/d | 0.16 | 0.14 | 0.21 | 0.14 | 0.14 | 0.14 | 0.14 | 0.21 | 0.14 | 0.21 |
| | Total volume used per household per day | L/p/d | 8.96 | 6.60 | 16.80 | 6.09 | 7.84 | 6.40 | 7.84 | 6.85 | 3.92 | 6.89 |
| Clothes washing (hand) | Number of clothes-washing sessions | wsh/d | 0.25 | 0.28 | 0.29 | 0.28 | 0.29 | 0.28 | 0.29 | 0.21 | 0.14 | 0.21 |
| | Volume of water used per wash per day | L/wsh/d | 19.25 | 15.43 | 19.72 | 8.94 | 20.01 | 10.00 | 19.72 | 16.61 | 12.65 | 23.24 |
| Vehicle washing | Number of vehicles washed per day | wsh/d | 2.00 | 2.00 | 1.43 | 1.41 | 1.43 | 1.43 | 1.71 | 1.71 | 4.00 | 2.00 |
| | Volume used per day | L/wsh/d | 11.53 | 10.25 | 9.8 | 8.70 | 9.6 | 9.04 | 10.9 | 9.78 | 12.0 | 11.12 |
| Cooking | Volume of water consumed in cooking | L/p/d | 10.83 | 14.15 | 9.79 | 11.16 | 9.57 | 11.14 | 10.87 | 13.60 | 15.22 | 17.80 |
| Drinking | Volume of water consumed for drinking | L/p/d | 4.38 | 3.78 | 4.9 | 3.48 | 4.3 | 3.56 | 4.3 | 3.85 | 4.6 | 3.94 |
| Garden | Volume of water consumed for garden | L/p/d | 0.0 | 9.18 | 0.0 | 7.14 | 0.0 | 7.50 | 0.0 | 9.15 | 0.0 | 11.00 |
| | Total water consumption | L/p/d | 120 | 89 | 109 | 64 | 106 | 70 | 125 | 92 | 132 | 111 |

Note: L/p/d = litre per person per day, L = litre, p = person, d = day, wsh = washes, min = minute, vol = volume, bt = bath, shw = shower, sec = second, brt = bathroom tap, tf = toilet flushing, lat = latrine, pf = pour flush, fl = flush, dws = dishwash, No./d = number per day.

The second largest water consumption was observed for bathing (bucket), i.e., 14%–16%, on average, for the rainy and 14%–19%, on average, for the dry months. The comparison of rainy and dry surveys showed that the number of households consuming higher than 23 L/p/d for bathing decreased from 52% in the rainy to 6% of households in the dry season. This can be due to the seasonality of water during the rainy months compared to the dry months. This is in accordance with the finding that developing countries generally use a much lower volume of water for bathing (5 to 15 L/p/d) [38]. In agreement with the rainy survey results for most of the other water end-uses, the analysis of dry season water consumption shows a decrease in the volume for hand wash basin taps in the surveyed households (Table 7). In terms of duration and flow rate of the use of hand wash basin taps, there is a slight decrease between the rainy and dry season (Table 7) across income groups. Then again, the number of uses decreases from 3.06 during the rainy months to 2.06 bathroom tap uses per day during the dry months, suggesting the impact of seasonality.

Table 7 shows no significant change in daily per capita water consumption for dishwashing. The number of times dishes are washed remains the same across all income groups, as dishwashing is done in a bowl of water and decreases slightly in the dry season. The average amount of water used in each toilet use type decreased slightly between both seasons. Consequently, the daily per capita water use for toilet flushing is slightly different between the rainy and dry season for all toilet types during the surveys. More households are using pit latrines in the dry season because of water scarcity. The number of times clothes washing is done per day increased from 0.25 during the rainy season to 0.28 washes per day during the dry months. The explanation for this could be the dusty weather. Approximately 46% of households tend to use more than 20 L/p/d for clothes washing in the rainy season, while 3% increase their consumption to more than 40 L/p/d in dry months.

The amount of water an average person would need to drink for a day is about 3 L/p/d, and it depends on the surrounding environment and weather conditions [38,65]. The estimate of the average per capita daily water consumption for the survey is given in Table 7. However, as the study area falls within the tropical climate, the analysis shows that the number of surveyed households which consume more than 3.5 L/p/d increases from 73% in the rainy season to 100% households in dry season.

The volume of each house washing session decreases from 8.96 L/p/d in the rainy season to 6.60 L/p/d during the dry season (Table 7). This may be due to physical and economic scarcity as a result of the change in rainfall patterns during the dry season [66]. Water availability during the seasons has an impact on the per capita water end uses.

*4.4. Limitations*

First, the sample has a higher percentage of middle-income households compared to the slum- and low-income households that does not reflect the general population of the study area. Secondly, citing and referencing previous research studies relevant to the study area are limited. Thirdly, the research was unable to assess each of the separate individual volumes (namely: Piped water and all multiple household water sources) of water used in the study area. These limitations would influence the overall average per capita water consumption and, therefore, be unable to determine the actual average daily per capita water provided by the Guma Valley service provider. Future studies should be designed to take into consideration the volume of water accessible by households from each service facility type. This would be necessary to increase water security and seasonal reliability.

**5. Conclusions**

This paper studied the determinants of per capita water consumption at the end-use level in a low- and middle-income urban city, Freetown. The impact of household characteristics (demographic, socio-economic and physical) on per capita water consumption was investigated. The significant finding is that insufficient water supply is predominant

in the city and very little or no research has been conducted to understand the factors affecting water scarcity and what coping mechanisms have been employed by residents. This research has been relevant to respond to the limited attention directed towards present research by studying 398 households of varying income levels, to extract information on household, water user habits and the intensity of indoor and outdoor water use activities. Furthermore, 20 statistical models, based on stepwise regression analysis, were developed to estimate daily per capita water consumption based on household socio-economic characteristics.

The results revealed that per capita water consumption in litres per day was positively correlated with family income and the number of containers used by households for water storage. However, it was significantly negatively affected by distance to water points and the time spent to fetch water and return home. The study establishes that a seasonal variation has a considerable impact on per capita water consumption. The average per capita water consumption varied from 151 L/p/d in the rainy season to 105 L/p/d in the dry season depending on the available multiple water sources to households. The study further revealed that piped water was extremely insufficient to meet the daily per capita water needs of the households.

Our findings indicate that the water service provider can barely serve its customers adequately, and some parts of the city receive no service at all. Therefore, volumetric water pricing is not an effective strategy to regulate per capita water consumption. However, it has generated the need to develop a policy to install more serving water points at short distances to reduce the long distance covered and queuing time.

The data sourced and analysed will serve as an input to the 3D Model Muse MOD-FLOW code for groundwater pumpage and abstraction in the study area. Future studies on domestic water consumption in the area should pay more attention to water usage habits and the different volumes of water used from the multiple household water sources by households.

**Author Contributions:** Conceptualization, S.I.A. and F.A.M.; methodology, S.I.A.; software, S.I.A.; validation, S.I.A., F.A.M. and D.B.; formal analysis, S.I.A.; investigation, S.I.A.; resources, S.I.A.; data curation, S.I.A.; writing—original draft preparation, S.I.A.; writing—review and editing, S.I.A., F.A.M.; D.B. supervision, F.A.M.; and D.B. funding acquisition, S.I.A. All authors have read and agreed to the published version of the manuscript.

**Funding:** This research was funded via a PhD scholarship from the Schlumberger Stitching Fund, Faculty for the Future, www.foundation.slb.com (accessed on 1 September 2014).

**Institutional Review Board Statement:** Not applicable.

**Informed Consent Statement:** Not applicable.

**Data Availability Statement:** The data presented in this study are available on request from the corresponding author.

**Acknowledgments:** The authors would like to thank the Schlumberger Stitching Fund, Faculty for the Future for financial support. The authors are very grateful to the students and staff members of Fourah Bay College, University of Sierra Leone for their support in all stages of this research investigation.

**Conflicts of Interest:** The authors declare no conflict of interest.

# Appendix A

**Table A1.** Models and coefficient of determination (R2) using multiple linear regression method (STEPWISE).

| | Model | R² | |
|---|---|---|---|
| | | Calibration Set | Validation Set |
| All investigated households | Model based on demographic characteristics of the household<br>$TW_w = 169.90 - 10.97 \times Nc_w - 20.25 \times N_{AFw} - 12.34 \times N_{AMw} + 18.58 \times E_{66-75w} - 23.81 \times E_{>76w}$ | 0.64 | 0.68 |
| | Model based on physical characteristics of the household<br>$TW_w = 169.52 - 1.60 \times N_{ROw} - 14.28 \times N_{HSw} + 2.14 \times A_w - 3.85 \times N_{FLw}$ | 0.69 | 0.75 |
| | Model based on water-use characteristics of the household<br>$TW_w = 107.25 + 0.88 \times S_{Hw} + 1.04 \times F_{Vw} + 0.02 \times T_{Sw} + 0.89 \times D_{Sw}$ | 0.65 | 0.70 |
| | Model based on all (demographic, physical and water use) characteristics of the household<br>$TW_w = 158.17 + 10.51 \times Nc_w + 8.65 \times N_{AMw} + 7.82 \times N_{AFw} + 17.82 \times E_{66-75w} + 13.92 \times E_{>76ww} - 2.53 \times N_{ROw} - 9.36 \times N_{HSw} - 0.74 \times A_w - 1.83 \times N_{FLw} + 0.52 \times S_{Hw} + 1.65 \times F_{Vw} - 4.56 \times T_{Sw} - 8.44 \times D_{Sw}$ | 0.75 | 0.77 |
| Informal settlement households | Model based on demographic characteristics of the household<br>$TW_S = 160.34 - 17.74 \times Nc_s - 19.51 \times N_{AMS} - 24.32 \times N_{AFS} - 22.18 \times E_{66-75S} - 24.47 \times E_{>76S}$ | 0.84 | 0.82 |
| | Model based on physical characteristics of the household<br>$TW_S = 173.06 - 17.76 \times N_{ROs} - 19.81 \times N_{HSs} - 0.74 \times A_s - 15.50 \times N_{FLs}$ | 0.80 | 0.89 |
| | Model based on water-use characteristics of the household<br>$TW_S = 172.26 + 0.52 \times S_{HS} + 4.47 \times F_{VS} - 0.94 \times T_{SS} + 0.62 \times D_{SS}$ | 0.86 | 0.83 |
| | Model based on all (demographic, physical and water use) characteristics of the household<br>$TW_S = 136.89 + 10.12 \times N_{Cs} + 8.76 \times N_{AMs} + 17.11 \times N_{AFs} - 13.71 \times E_{66-75s} + 20.148 \times E_{>76ss} + 17.87 \times N_{ROs} + 13.09 \times N_{HSs} - 0.32 \times A_s + 3.24 \times N_{FLs} - 0.87 \times S_{Hs} + 6.80 \times F_{Vs} + 0.52 \times T_{Ss} - 0.16 \times D_{Ss}$ | 0.93 | 0.92 |
| Low-income households | Model based on demographic characteristics of the household<br>$TW_l = 174.63 - 12.61 \times N_{Cl} - 15.46 \times N_{AMl} - 23.14 \times N_{AFl} - 13.29 \times E_{66-75l} - 24.91 \times E_{>76l}$ | 0.74 | 0.78 |
| | Model based on physical characteristics of the household<br>$TW_l = 154.96 - 0.58 \times N_{ROl} - 14.49 \times N_{HSl} + 0.52 \times A_l - 3.17 \times N_{FLl}$ | 0.82 | 0.88 |
| | Model based on water-use characteristics of the household<br>$TW_l = 110.90 + 1.70 \times S_{Hl} + 3.69 \times FV_l - 0.73 \times T_{SS} - 1.96 \times D_{SS}$ | 0.78 | 0.83 |
| | Model based on all (demographic, physical and water use) characteristics of the household<br>$TW_l = 143.17 + 28.07 \times N_{Cl} + 19.57 \times N_{AMl} + 6.28 \times N_{AFl} - 10.75 \times E_{66-75l} + 21.72 \times E_{>76ll} + 11.78 \times N_{ROll} + 23.03 \times N_{HSl} - 0.63 \times A_l + 13.50 \times N_{FLl} + 1.34 \times S_{Hl} + 2.00 \times F_{Vl} + 0.26 \times T_{Sl} - 2.75 \times D_{Sl}$ | 0.86 | 0.92 |

<div align="center">

Table A1. *Cont.*

</div>

| Model | | $R^2$ | |
|---|---|---|---|
| | | **Calibration Set** | **Validation Set** |
| Middle-income households | Model based on demographic characteristics of the household<br>$TW_m = 176.00 - 9.69 \times N_{Cm} - 17.36 \times N_{AMm} - 19.78 \times N_{AFm} - 17.83 \times E_{66-75m} - 20.72 \times E_{>76m}$ | 0.74 | 0.76 |
| | Model based on physical characteristics of the household<br>$TW_m = 186.65 + 0.67 \times N_{ROm} - 15.00 \times N_{HSm} - 0.54 \times A_m - 3.57 \times N_{FLm}$ | 0.81 | 0.84 |
| | Model based on water-use characteristics of the household<br>$TW_m = 98.87 + 0.52 \times S_{Hm} + 1.66 \times F_{Vm} - \times 1.77\, T_{Sm} + 0.72 \times D_{Sm}$ | 0.80 | 0.82 |
| | Model based on all (demographic, physical and water use) characteristics of the household<br>$TW_m = 141.57 - 3.43 \times N_{Cm} - 4.59 \times N_{AMm} - 10.05 \times N_{AFm} - 5.32 \times E_{66-75m} - 16.73\, E_{>76mm} + 2.80 \times N_{ROm} - 3.71 \times N_{HSm} - 0.69 \times A_m - 3.36 \times N_{FLm} + 1.56 \times S_{Hm} + 1.92 \times F_{Vm} - 0.21 \times T_{Sm} - 2.61 \times D_{Sm}$ | 0.73 | 0.85 |
| High-income households | Model based on demographic characteristics of the household<br>$TW_h = 163.4 - 8.09 \times N_{Ch} - 16.42 \times N_{AMh} - 19.60 \times N_{AFh} - 19.61 \times E_{66-75h} - 4.83 \times E_{>76h}$ | 0.84 | 0.81 |
| | Model based on physical characteristics of the household<br>$TW_h = 251.50 - 3.67 \times N_{ROh} - 26.68 \times N_{HSh} + 0.78 \times A_h - 5.64 \times N_{FLh}$ | 0.83 | 0.87 |
| | Model based on water-use characteristics of the household<br>$TW_h = 113.72 + 0.95 \times S_{Hh} + 1.99 \times F_{Vh} - 0.87 \times T_{Sh} - 6.38 \times D_{Sh}$ | 0.82 | 0.77 |
| | Model based on all (demographic, physical and water use) characteristics of the household<br>$TW_h = 270.81 - 28.97 \times N_{Ch} - 24.71 \times N_{AMh} - 33.14 \times N_{AFh} - 56.93\, E_{66-75h} - 13.63 \times E_{>76h} - 3.9 \times N_{ROh} \pm 26.31 \times N_{HSh} - 0.14 \times A_h - 15.46 \times N_{FLh} - 0.31 \times S_{Hh} + 2.75 \times F_{Vh} + 0.41 \times T_{Sh} + 4.71 \times D_{Sh}$ | 0.90 | 0.95 |

Notes: * TW = daily per capita water consumption (L/p/d), A = total household floor area (m²), w = whole sample, $N_C$ = number of children in the household, $N_{FL}$ = number of floors in the household, s = slum-income, household, $N_{AF}$ = number of adult females in the household, $S_H$ = shower volume (L), l = low-income households, $N_{AM}$ = number of adult males in the household, $F_V$ = flushing volume (L), m = middle-income, $E_{66-75}$ = number of elders 66–75 years in the household, $T_S$ = time spent to fetch water (L), h = high-income households, $E_{>76}$ = number of elders >76 years in the household, $D_S$ = distance to water point (m), $N_{RO}$ = number of rooms in the household, $N_{HS}$ = number of occupants in the household, m = middle income household.

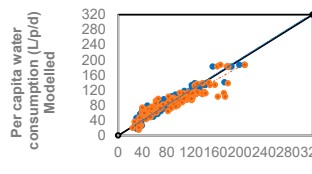
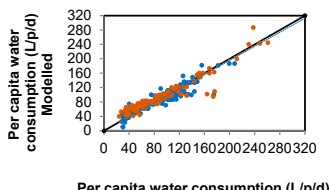
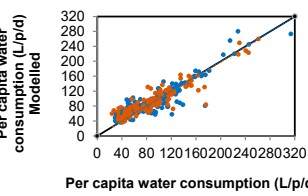
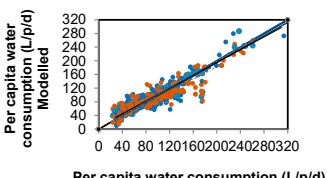

a. All investigated households based on demographic characteristics

b. All investigated households based on physical characteristics

c. All investigated households based on water use characteristics

d. All investigated households based on demographic, physical and water use characteristics

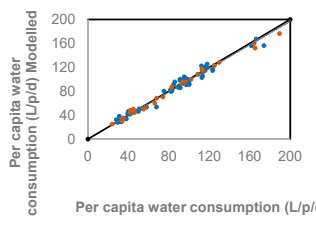
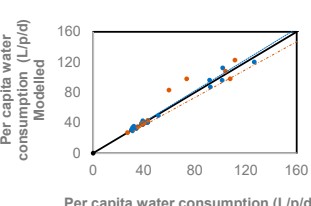
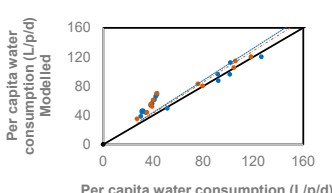
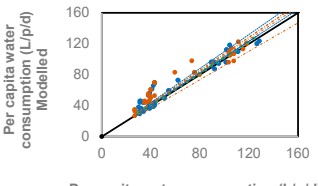

e. Informal slum households based on demographic characteristics

f. Informal slum households based on physical characteristics

g. Informal slum households based on water use characteristics

h. Informal slum households based on demographic, physical and water use characteristics

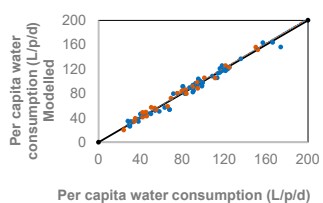
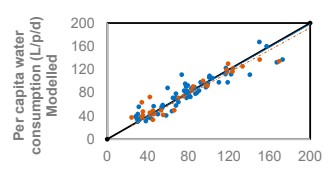
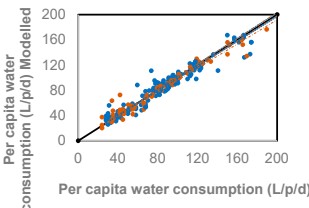

i. Low-income households based on demographic characteristics

j. Low-income households based on physical characteristics

k. Low-income households based on water use characteristics

l. Low-income households based on demographics, physical and water use characteristics

**Figure A1.** *Cont.*

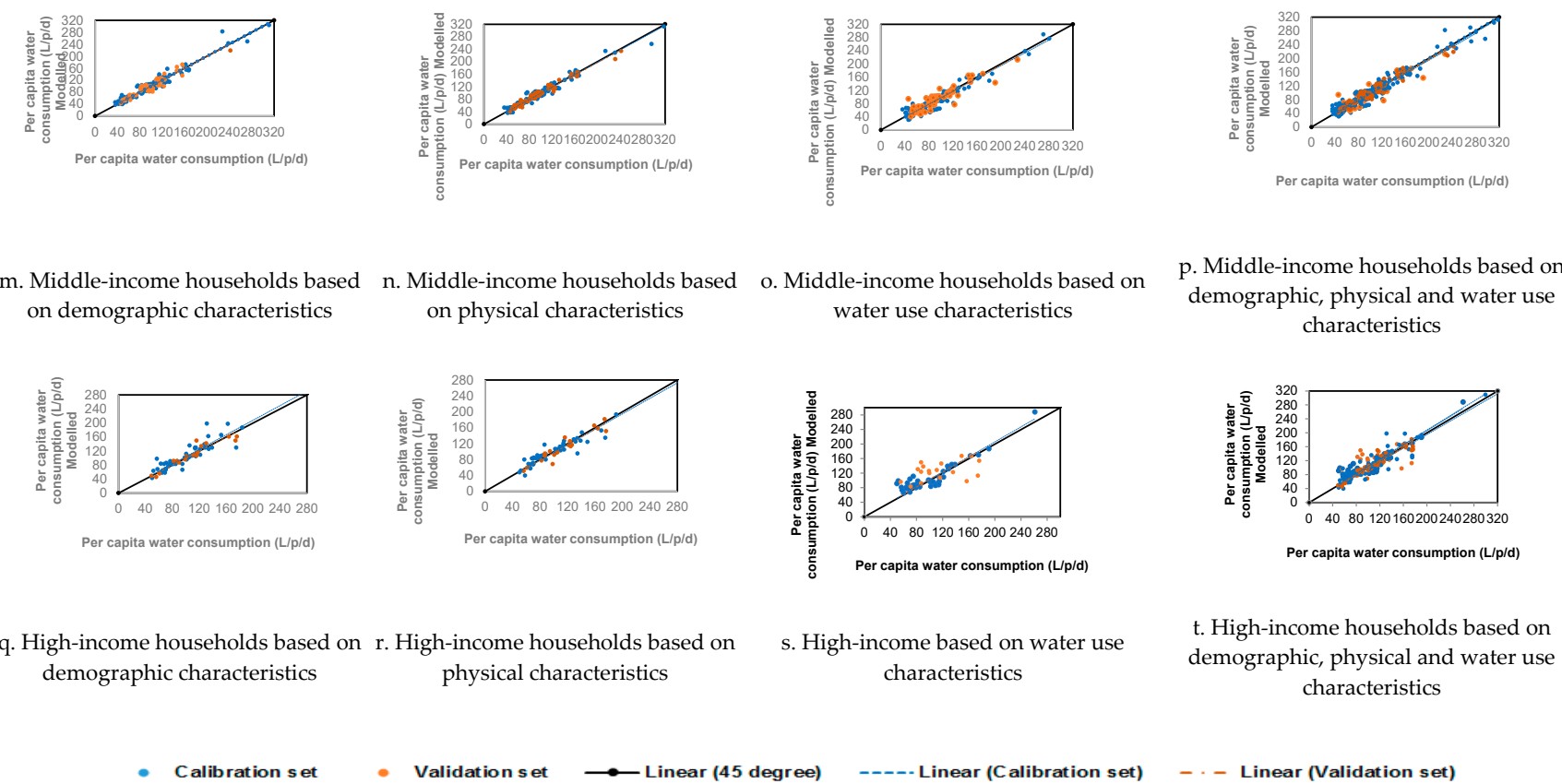

**Figure A1.** Relationship between actual and predicted daily per capita water consumption using linear regression stepwise method.

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
