# Peer review of "Seasonal Variation of Rainy and Dry Season Per Capita Water Consumption in Freetown City Sierra Leone"

_water, doi:10.3390/w13040499_

Round 1

Reviewer 1 Report

A full picture of the questionnaire does not provide. In addition, I cannot know how total and end-use consumption was calculated from the questionnaire.

Some figures are not well-described.

Some findings are not new and unique. For example, the relationship between per capita consumption and a number of occupants in the household is already suggested in existing studies. 

Fig 2

Please explain the detail of each water source. For example, I cannot understand the difference between vender and bowser. In addition, the figure is very hard to understand.

Captions of Figure 3 and 4 are incomplete.

Table 1 There should be more than just mean value, but also variance.

169, 183 You have to make chapter.

The sentence is incomplete.

Figure 5 No caption and the figure is wrong.

Figure 6 Why is the width of the histogram at 8? Figure 9, too.

Table 3 How to know the flow rate?

Table 4 Why water use characteristics are used as a variable in the analysis of total usage?

Table 6 Please delete **.

Author Response

Dear Reviewer,

Thank you for your time and feedback to improve on the manuscript.

Manuscript reference number: water-1008138

Reviewer #1:

A full picture of the questionnaire does not provide. In addition, I cannot know how total and end-use consumption was calculated from the questionnaire.

Response: Thank you for your clarification.  The questionnaire was detailed in estimating each water end use activity that water is used for, then further questions were asked on how much water was used for each activity namely: how much water do you use when you take a shower? How long do you take a shower for? What is the flow rate of the shower? How much water is used when you take a bucket bath? Wash a load of clothes? When you brush your teeth etc. As this questionnaire was in English and difficult for all members of the household to complete, university students of the households were targeted to respond to the questions on behalf of the family. The university lecturers facilitated the selection of students and training process to measure near accurate readings before the survey exercise.   The questions included the number of times activities that use water were taken, taking a bucket bath, water to clean-up after latrine use, dishwashing and clothes washing. They were asked to record the number of times each activity was done, the length of time for each activity that required running taps, the amount of water used for each activity. The households were asked what their collection and storage water containers were. Based on their responses, the total water and individual water end use were calculated.

We believe a high level summary on the survey characteristics, already provided in the manuscript is sufficient. However, we can add the above mentioned text in the manuscript or provide the questionnaire as a supplementary material, if the reviewer wishes us to do so. 

Some figures are not well-described.

Thank you for your observations these have been amended with best description. For example Figure 2 has been revised in a tabular form that details the different service facility in relation to household percentage access, additional text has been added for (Lines 104 - 108), (Lines 122-128), and Figures 3 and 4 etc  to further enhance clarity and associated message.

Some findings are not new and unique. For example, the relationship between per capita consumption and a number of occupants in the household is already suggested in existing studies. 

We accept that the influence of the household size on per capita consumption is well documented for the developed countries. However, an evidence base for the developing countries, in particular for Freetown, is rather sparse. The present study helps to provide some additional insights to determine additional factors that are likely to affect per capita consumption in contexts similar to Freetown. Related discussion can be found in the paper (e.g.  Section 3).

Fig 2 Please explain the detail of each water source. For example, I cannot understand the difference between vender and bowser. In addition, the figure is very hard to understand.

Thank you! The study area has critical water supply issues from the service provider. Tap water is the primary source for households where it exists. Tap water is rationed on alternate days for the different neighbourhoods. Taps are closed even in the rainy season and this gets worst in the dry season due to reduction in surface water which replenishes the dam.  Therefore, residents must find their own source of water to compliments their primary source. Water service is also a business for most people, so some people have engaged in water business such as selling water from water bowser or truck and small scale water business from vendors (hawkers) using pushcarts. Based on family income and accessibility, households find the suitable water sources for their needs.  Small scale vendors sell water loaded in jerry can (22 litres) container and water from tanker truck at their door step.  These are further described in the revised paper ( Lines 126 to 140)  We have now deleted Figure 2 and associated discussion.

Captions of Figure 3 and 4 are incomplete.

Captions have been made visible (Figure 2, 3, 4 etc)

Table 1 There should be more than just mean value, but also variance.

Variance parameter is included.

169, 183 You have to make chapter.

Thank you. Sub headings has been added.

  1. Distance to the sources of water inserted, lines 169 and 183.
  2. Time spent to water sources and return home

The sentence is incomplete.

Thank you. The sentence has been revised and completed. 

Figure 5 No caption and the figure is wrong.

Thank you! The caption has been made visible and properly described.

Figure 6 Why is the width of the histogram at 8? Figure 9, too.

The figure at that width present the analysed data in distinct comparison to the data.

Table 3 How to know the flow rate?

The lecturers facilitated the process. Respondents were taught and trained on how to measure flow rate of the running tap using a stop water and measuring container. First a container was used to measure the flow rate for 1 minute and the amount of water was measured. Then the frequency per day was noted. The flow rate was estimated by multiplying the duration of tap flow x frequency x volume of water from tap over the period. This is now explained (Lines 93 to 96).

Table 4 (now Table 5) Why water use characteristics are used as a variable in the analysis of total usage?

Thank you for your observation. Because water supply is grossly inadequate and unreliable, households have to search for available source and these have increased factors that affect it accessibility and affordability. Water use characteristics (shower volume, toilet flushing volume, time spent to fetch water, distance to water source, collection container size) have to be considered in the total analysis usage because these are factors that impact on the actual volume use for the different water end-use. Additionally, these provide insights into water user behaviour and can be used to develop water consumption models to predict water demand for contexts similar to Freetown.

Table 6 Please delete**.

Table 6 (now Table 7) Asterisk* has been deleted.

Reviewer 2 Report

This manuscript contains a lot of useful information on self-reported water use in Freetown, Sierra Leone. Data on urban water use in developing countries, with data on both volume and type of use, is rare and makes a nice contribution to the literature. The modeling results could also be useful although there’s so much presented that it’s a little overwhelming to interpret.  

The lit review is okay, but it seems to neglect the last 3 years of literature (I assume this manuscript draft was built off a slightly older work). For example, it completely neglects to cite any of the HWISE articles that have been published in a last couple years (https://hwise-rcn.org/outcomes/publications/). And there are some cases where obscure references are used when other highly cited works would make the case and lend more credibility to the authors; for example, the very first reference is to an obscure paper on seasonal water use in Andorra.

It doesn’t seem like the authors are familiar with the subdiscipline of research in global WASH typically termed “multiple household water sources and uses (MWSU)”; this is somewhat understandable as MWSU has emerged as a distinct area of inquiry only in the past 3-4 years. I refer the authors to two brief 2019 overviews of the topic (both open access) that have a variety of citations that the authors could use: https://www.nature.com/articles/s41545-019-0031-4, https://waterinstitute.unc.edu/files/2019/07/wash-policy-digest-11.pdf. And I think that framing it in those terms would generate more interest.

My main concern is a strange statement in the Methods section that may simply have been written in an unclear way. Lines 101-103 state: “The investigated households were categorized into four household income groups; informal slum settlement, low, middle and high-income households. The classification for the different income groups was based on the access to pipe borne water and payments of water tariffs, defined by the Local Governance Act [21].” Perhaps I’m misinterpreting this, but it seems to be saying that the four groups were defined based on piped water access and payments of water tariffs. I can’t see how one can define the independent variable that is used to stratify data based on access to/payment for the main dependent variable (water use), and then analyze differences between the groups based on the their use of the dependent variable. Hopefully it’s just a poorly worded explanation or I’m missing something. But I’m concerned that it invalidates all comparisons between the income groups.

Detailed Comments:

Abstract:

Line 9 – Use developing countries or LMIC. No one in development uses “third world.” Even if you think it’s the most accurate term, it will alienate half your audience.

Line 16: Maybe “behaviour around indoor…”

Line 23: Move “increases with income before the volumes if you want to use “respectively” at the end.

Line 24: The model was “trained on the data”, right, not the “data has been trained” (This is present elsewhere. Perhaps there is some nuance to your methodology that I’m missing.)

Line 28-30: Showering and bathing are usually consolidated into a single category, so this finding is even more striking. You might want to rephrase it to draw attention to this. However, it’s almost inevitable that with high latrine/pour-flush toilet use that water use will be low for toilet flushing.

Introduction:

Line 35-36: Be more careful when choosing your first couple references. It can hurt your credibility with readers to not cite the most relevant and recent studies.

I think you should refer more clearly to your volume data being “self-reported” by the households.

Methods:

Line 110: Obviously if you fit 20 different models, some of them will fit the data. It’s unclear why you used so many. Perhaps you could emphasize that not all these variables are available in all cases and thus other researchers might want to use/build off one of your models that includes the variables to which they have access.

Results:

Figure 2: This is an incredible breadth of water sources used by your sample. You should really calculate how many sources these homes used on average and plot that. Similar to what you find in the MWSU literature referred to above.

I would also recommend that you present these values as % of HHs in that group. The slum data are hard to evaluate because the sample is so small.

Also, these aren’t all “Improved” sources as stated in the caption (river, unprotected well, etc.)

Line 157: “Water consumption per capita” right?

Line 178: This standard has been 30 min, not 100 meters.

Figure 5: This is really striking the time people are queueing for water. But how can it be that only 21% of the sample wait 0-30 min for water when at least 33% have a source on plot? Is it because people use their piped water (or well) on their plot, but they still queue for higher quality water for consumption? If so, that’s very interesting and shows a high value (in time spent) placed on water quality.

Figure 6: Really interesting that the vast majority of HHs use 83-121 l/cap/day. That’s a pretty narrow range given the different sources and wait times.

Table 3: Some of the References cited on the right (label that column BTW) are in different units than those you used. For example, what should be “persons in HH taking bath per day” seems to be compared to a volume metric from Gleick.

Line 310: Seems to imply that an automatic clothes washer uses less water than washing clothes in a bowl (?)

Line 343-344: This statement is not correct if you’re using “Improved” here as you have been throughout the paper. In 2015, the total global number without “Improved” DW was 640 million or so according to JMP. Yes, rural-urban migration increased global population enough that even with gains in % Improved coverage in urban areas, there could possibly be a slight increase in the total number of urban residents without access. But there is no way it was 844 million only in urban areas in 2015. You can’t switch to using “Safely Managed” as the criterion halfway through the manuscript if that’s where that number came from. And if you’re citing one of the other safety oriented analyses of Improved from the MDG period (e.g., Onda et al., 2012, Bain et al., etc.) you need to cite it.

Line 357: Incorrect use of semicolon.

Table 5 is really hard to read/interpret.

Line 430+: There is a really good paper, Foster and Hope, 2017 from WRR that should be cited here. It has great data on seasonal water use in a few African countries.

Figure 9: Really nice figure, very striking difference by season.

Table 6: Can you add a % difference column? I realize you’ll get NA for Garden Watering but it would be useful.

Given that you used university students going back to their homes, you have to concede that it’s possible you have a systematically biased sample. It’s also unclear whether the student was counted as living in the home in the demographic and HH data, or whether the students mostly lived at/near the university.

I was expecting a Limitations subsection at the end. I think it would be appropriate. 

Author Response

Dear Reviewer,

Thank you for your time and feedback to improve this manuscript. Your effort is highly appreciated.

Reviewer #2:

This manuscript contains a lot of useful information on self-reported water use in Freetown, Sierra Leone. Data on urban water use in developing countries, with data on both volume and type of use, is rare and makes a nice contribution to the literature. The modelling results could also be useful although there’s so much presented that it’s a little overwhelming to interpret.  

Thank you. We appreciate reviewer for rightly spotting the data scarcity for developing countries contexts. This paper attempts to provide some evidence base focused on Freetown.

The modelling results could also be useful although there’s so much presented that is a little overwhelming to interpret.

We take the reviewer’s point and have now moved the regression models to the supplementary material. In the manuscript, we have now only presented information as an essay to follow set of diagrams showing the prediction performance of the developed models.

The lit review is okay, but it seems to neglect the last 3 years of literature (I assume this manuscript draft was built off a slightly older work). For example, it completely neglects to cite any of the HWISE articles that have been published in a last couple years (https://hwise-rcn.org/outcomes/publications/). And there are some cases where obscure references are used when other highly cited works would make the case and lend more credibility to the authors; for example, the very first reference is to an obscure paper on seasonal water use in Andorra.

Thank you for your observation and the recommendation of the https://hwise- rcn.org/outcomes/publications. These recent literature have been referenced in the overall review (Lines 75, 81, 129 and 355); We have also deleted the following dated articles

Griffin and Chang, (1991) Fan L, H. van Zyl, J. (2007) have been deleted.

It doesn’t seem like the authors are familiar with the subdiscipline of research in global WASH typically termed “multiple household water sources and uses (MWSU)”; this is somewhat understandable as MWSU has emerged as a distinct area of inquiry only in the past 3-4 years. I refer the authors to two brief 2019 overviews of the topic (both open access) that have a variety of citations that the authors could use: https://www.nature.com/articles/s41545-019-0031-4, https://waterinstitute.unc.edu/files/2019/07/wash-policy-digest-11.pdf. And I think that framing it in those terms would generate more interest.

Thank you so much for your reference to the above mentioned articles. We have now integrated these articles in the revised version (Lines 75, 355 and 548).

My main concern is a strange statement in the Methods section that may simply have been written in an unclear way. Lines 101-103 state: “The investigated households were categorized into four household income groups; informal slum settlement, low, middle and high-income households. The classification for the different income groups was based on the access to pipe borne water and payments of water tariffs, defined by the Local Governance Act [21].” Perhaps I’m misinterpreting this, but it seems to be saying that the four groups were defined based on piped water access and payments of water tariffs. I can’t see how one can define the independent variable that is used to stratify data based on access to/payment for the main dependent variable (water use), and then analyze differences between the groups based on the their use of the dependent variable. Hopefully it’s just a poorly worded explanation or I’m missing something. But I’m concerned that it invalidates all comparisons between the income groups.

Thank you for your question. More explanation has been provided between lines 85 and 94.

Freetown is divided into neighbourhood wards that defines status of household income group. The slum communities (deprived of piped network) are mostly along the coast. The central to East II wards are densely populated and concentrated mainly by low income households. Households in far East of Freetown (East III) have some facilities, while the western wards (West I – III) have metred piped and classified as better off areas.  

However, Freetown City Council & Guma Valley has flat tariff for houses without metered connection. Flat rates are distinguished between the different household accessibility to piped network. This fee is paid to the City Council as part of the house tax, annually by the user. The classification into the different groups is supported in the Supply Water Framework document of “Guma Valley Water Company, Freetown Republic of Sierra Leone AND SANITATION FRAMEWORK,” no. March, 2008.

Detailed Comments:

Abstract:

Line 9 – Use developing countries or LMIC. No one in development uses “third world.” Even if you think it’s the most accurate term, it will alienate half your audience.

Thank you and revision made

Line 16: Maybe “behaviour around indoor…”

Thank you, suggestion implemented

Line 23: Move “increases with income before the volumes if you want to use “respectively” at the end.

Thank you, line 23 adjusted. 

Line 24: The model was “trained on the data”, right, not the “data has been trained” (This is present elsewhere. Perhaps there is some nuance to your methodology that I’m missing.)

The model was trained using 70% of the collected 70% of data. The remaining 30% of the data was used for validation purposes. This has been now corrected (Lines 24, 117, 437-439).

Line 28-30: Showering and bathing are usually consolidated into a single category, so this finding is even more striking. You might want to rephrase it to draw attention to this. However, it’s almost inevitable that with high latrine/pour-flush toilet use that water use will be low for toilet flushing.

Thank you for your observations, some respondents who indicated taking a shower, also do have a bucket bath when the taps are dry. Households without piped water only take bucket baths. The volume used for showering and a bucket bathing is not uniform across the different income groups. This was why showering and bucket bathing was not put in into a single category. This has been further clarified in the revised version (Line 291 to Line 305).

Introduction:

Line 35-36: Be more careful when choosing your first couple references. It can hurt your credibility with readers to not cite the most relevant and recent studies.

The dated and the references at the beginning have now been replaced with recent and most relevant ones (Line 36, 46, 75, 132, etc). Thank you for the valuable suggestion.

I think you should refer more clearly to your volume data being “self-reported” by the households.

Thank you for your observations. Volume data included here is self-reported measurements. The suggestions have been revised and taken into consideration (Line 70 to 92).

 Methods:

Line 110: Obviously if you fit 20 different models, some of them will fit the data. It’s unclear why you used so many. Perhaps you could emphasize that not all these variables are available in all cases and thus other researchers might want to use/build off one of your models that includes the variables to which they have access.

Have now incorporated this suggestion. Not all these variables are available in all cases. Since in this study, we managed to collect data on a range of variables, it was decided to explore fully the influence of all variables on consumption. This resulted in the development of over 20 models. These can help any future researchers to adapt to the most suitable model sets depending on data availability. Lines 437 to 498.

Results:

Figure 2: This is an incredible breadth of water sources used by your sample. You should really calculate how many sources these homes used on average and plot that. Similar to what you find in the MWSU literature referred to above.

Thank you for your observation. Working on your request to plot average number of sources by each household as in Elliot et al.  Nearly all of the households used at least three of the multiple water sources. Majority have a different source (Bottled or packaged) for drinking from other sources. Some households indicated more than three sources depending on accessibility, distance, and kind of domestic needs. The primary, secondary and tertiary sources were identified for the different income groups.

I would also recommend that you present these values as % of HHs in that group. The slum data are hard to evaluate because the sample is so small.

Also, these aren’t all “Improved” sources as stated in the caption (river, unprotected well, etc.)

Thank you for your observation. This figure has been improved to Table 1 and data presented as percentage of seasonal household access line 145.

 Line 157: “Water consumption per capita” right?

Thank you for your observation. Yes! per capita water consumption

Line 178: This standard has been 30 min, not 100 meters.

Thank you for your observation. Sentence revised.

Figure 5: This is really striking the time people are queueing for water. But how can it be that only 21% of the sample wait 0-30 min for water when at least 33% have a source on plot? Is it because people use their piped water (or well) on their plot, but they still queue for higher quality water for consumption? If so, that’s very interesting and shows a high value (in time spent) placed on water quality.

Thank you for your observation in line 59 to 62. Tap water is rationed not only on days but the time allocated to the different areas or households and is less than 24 hours. The water pressure and flow rate are low and most people with taps have indicated that they have to store water in their storage containers. In some communal compound houses with tap water, residents (and neighbours) queue in compounds to fetch water. Most of the private wells are hand dug, and not properly sited to this affecting the pumping and replenish time which affect withdrawal time.

Figure 6: Really interesting that the vast majority of HHs use 83-121 l/cap/day. That’s a pretty narrow range given the different sources and wait times.

Thank you for your observations. Households have to queue and fetch water within the tap running times. Households have to develop and time rooster at public multiple household water use sites at public standpipes or from neighbours and equitable water collection time is allocated. This is further discussed in the revised version (Lines 143 to 144).

This figure represents the typical total volume of water used per day, considering all the water end uses (clothes washing, housecleaning, vehicle washing, drinking, cooking etc) are done in a day. It is possible this amount could be lower in a day if clothes washing, housecleaning or vehicle washing are all not done on the same day.

Table 3: Some of the References cited on the right (label that column BTW) are in different units than those you used. For example, what should be “persons in HH taking bath per day” seems to be compared to a volume metric from Gleick.

Thank you for your suggestion and observation. The volume is referenced from P. Gleick’s 1996 paper on page 88 Table 9 ‘Recommended basic water requirements which are in litres per capita per day. The references citied in the column are in litres as per the different end use. The column compared past studies and the references cited have been updated to reflect the unit used in the study lines 357 to 360.

Line 310: Seems to imply that an automatic clothes washer uses less water than washing clothes in a bowl (?)

Thank you for your observation. This has been revised in lines 362 – 370. Washing clothes by machine depends on the technology and can range from 50 litres per wash to above depending on the age of the appliance.

In this study, none of the respondent recorded use of a washing machine and such implication is not sanctioned here. In the survey, because of water scarcity and accessibility, households are using the best water value for their end uses.

Line 343-344: This statement is not correct if you’re using “Improved” here as you have been throughout the paper. In 2015, the total global number without “Improved” DW was 640 million or so according to JMP. Yes, rural-urban migration increased global population enough that even with gains in % Improved coverage in urban areas, there could possibly be a slight increase in the total number of urban residents without access. But there is no way it was 844 million only in urban areas in 2015. You can’t switch to using “Safely Managed” as the criterion halfway through the manuscript if that’s where that number came from. And if you’re citing one of the other safety oriented analyses of Improved from the MDG period (e.g., Onda et al., 2012, Bain et al., etc.) you need to cite it.

Thank you for your observation, this is noted. The statement has been revised in lines 398 - 404.

The sentence was referenced to the basic service type which read ‘844 million people still lacked even a basic drinking water service’ from the WHO/UNICEF, “Progress on Drinking Water , Sanitation and Hygiene. Launch version July 12 Main report Progress on Drinking Water , Sanitation and Hygiene.,” WHO Libr. Cat. Publ. Data, p. 140, 2017.

Line 357: Incorrect use of semicolon.

Thank you for your observation. Corrected.

Table 5 is really hard to read/interpret.

Using the calibration set of data (70 % for all the 398 households) four nonlinear regression models are developed as a function of demographic, physical, water use and all characteristics and (Model 1, 2, 3, and 4 in Table 6, respectively).  Equally four mathematical models were developed for each income group (slum, low, middle and high). The models have been selected due to achieving the highest coefficient of determination (R2) value. This has been moved to the supplementary material section and the related text has been further clarified as explained previously,

Line 430+: There is a really good paper, Foster and Hope, 2017 from WRR that should be cited here. It has great data on seasonal water use in a few African countries.

Thank you for your support! The Reference has been used. Line 81

Figure 9: Really nice figure, very striking difference by season.

Thank you for your observation!

Table 6: Can you add a % difference column? I realize you’ll get NA for Garden Watering but it would be useful.

Thank you! Table 6 now Table 7 has been modified with a %age difference column.

Given that you used university students going back to their homes, you have to concede that it’s possible you have a systematically biased sample. It’s also unclear whether the student was counted as living in the home in the demographic and HH data, or whether the students mostly lived at/near the university.

Thank you for your observations. This has been reflected in lines 93 to 97, 112-113. The students were not based on campus during the survey and they were identified based on their different locations in the study area. Rehabilitation work is ongoing at the campus and all students who took part in the survey were residing at their respective households.

I was expecting a Limitations subsection at the end. I think it would be appropriate. 

Thank you for your reviews, Limitation sub-section is included in lines 584 - 593.

Reviewer 3 Report

Abstract

Line no-24: What is the meaning of 'data have been trained to develop......'?

Line no-28: Can you combine the use for showering and bathing?

Introduction

Line no-44: Please clarify the availability of water or freshwater?

Line no-54, 315, 319, 469, and 522: Reference at the beginning of the sentence like [3], [11]–[15] is not appropriate. Please cite it instead of references.

Line no-62: Could you please give some data or trend statistics to believe these facts?

Materials and Methods

Line no-71: Could you please add the source in the second sentence?

Line no-81: Is this the Freetown shreds of evidence? If not, please search for some evidence that showed the water scarcity situation in Freetown.

Line no-94: How can you claim that your questionnaire is standard? Is this a self-administered questionnaire or face to face questionnaire? Who are your respondents? How you assure the reliability of water quantity measurement??

Line no-102: Please elaborate basis for categorizing HH into 4 income groups.

Result and Discussion

Line no-129: In Figure 2, could you please show the water sources in different ways - e.g. improved/unimproved source; pipe connection/unpiped connection, etc?

Line no-138-141: What is the significance of this information in your research?

Line no-144: Please complete this sentence, not understandable.

Line no-154: The relation you showed in this sentence is obvious, no need to establish. Could you please show it in different ways?

LIne no-165: " this survey’s targeted university students to respond on behalf of their households," should be mention in the method section. This group may have better income and better access to water than other residents. How can you claim it as a representative group from the community?

Line no- 179: This may be 'Total HH percentage' not 'Total % distance'.

Line no-197: "93 litres per capita per day (l/p/d)". How you measured the total averaged amount of water used? How do you clarify the reliability? Please write it in the method section.

Line no-216 & 294: what is the meaning of the full sample? It may be a total sample or please describe it in the methods section.

Line no-258: In the sentence "The smallest household size (2 persons) have the highest water consumption per...." How can you justify small HH sizes have the highest per capita water consumption?

Line no-305: What is the heading of the last column of Table 3.

Line no-347: In the sentence " average per capita drinking consumption is 4 liters per day" - What about the ranges of data? Could you please mention the seasonal variation in Freetown?

Line no-420: letter 'h' is missing in 'hig'.

Line no-423: On what basis, you calculated the TW average volume of water use? Any references for calculation? Please mention this in the methods section.

Line no-436: Is this the same survey or a different survey for your comparison?

Line no-453: Reference 51 of the sentence - Please complete this sentence, not understandable.

Line no-504: "who sites that.." maybe "WHO sites that..." or any other suggestion?

Line no-510: "2.06 brt/p/d" - what is brt?? Please write its full form.

Conclusion

Line no-532: Your conclusion seems abstract, very long, and not clear. Please rewrite it

Reference

Line no-583: Could you please add DOI on the references if applicable e.g. Referernce-1 (DOI 10.3390/w10030321 and Reference-3 (DOI 10.1029/2012wr012398)?

Line no-585: Please mention the website or ISBN of reference 2.

Please check your all references from 1-58.

Author Response

Dear Reviewer,

Thank you so much for taking the time to review and for giving us the opportunity to improve on the manuscript. 

Reviewer # 3

Abstract

Line no-24: What is the meaning of 'data have been trained to develop......'?

Well spotted. Thanks. This has been corrected. It was models trained using the collected data. Lines 24-25, 116 - 118

Line no-28: Can you combine the use for showering and bathing?

We think it is best to leave them as separate water end-uses. Only 46% of the surveyed households do take a shower and this is replaced by a bucket bath when taps are closed. With bucket baths, all respondents indicated having one.

Introduction

Line no-44: Please clarify the availability of water or freshwater?

Thank you for your observation. Clarified! Line 44

Line no-54, 315, 319, 469, and 522: Reference at the beginning of the sentence like [3], [11]–[15] is not appropriate. Please cite it instead of references.

Thank you for your observation. These have been revised in Line no-54, 315, 319, 469, and 522.

Line no-62: Could you please give some data or trend statistics to believe these facts?

Thank you for your suggestion. Correction has been added in line 62

Materials and Methods

Line no-71: Could you please add the source in the second sentence?

Thank you for your observation. The source has been added. Line 71

Line no-81: Is this the Freetown shreds of evidence? If not, please search for some evidence that showed the water scarcity situation in Freetown.

This is actually Freetown evidence and has been supported with a water scarcity issue reference.  

Line no-94: How can you claim that your questionnaire is standard? Is this a self-administered questionnaire or face to face questionnaire? Who are your respondents? How you assure the reliability of water quantity measurement??

The questionnaire was self-administered therefore offered no interviewer’ bias. The questions were easy and they give detailed information about any small or large household.  They were distributed anonymously to produce more accurate responses to the questions asked. They had pre-categorized responses. The students/respondents were living at their respective houses across the study area. They completed the questionnaire on behalf of their households. These students were trained on how to respond to the questionnaire as well as how to measure and estimate flow rates and volume of the different water used on a daily basis.

Line no-102: Please elaborate basis for categorizing HH into 4 income groups.

In the 2004 census data, it reported that 65% of habitats are supplied by the GVWC. This situation has not improved and coverage differs across the study area. In the western part of the city piped water coverage is from 89% and decreases to 42% in the eastern part. Areas not supplied by the GVWC network have relied on protected and unprotected wells, gravity, streams, and water vendors for water. There are also areas with limited or no service at all.

Settlement, location and house structure also influence water accessibility and affordability. Settlement areas also reflect income status and access to basic facilities. Before a household is charged or billed for water rates, an assessment on the house structure and location is conducted by the Freetown City Council, who then sends the assessment for the charge to the Guma Valley.

Based on the location of the household, they are classified as deprived communities, densely populated, etc to better-off areas with metered connections as in Figure 1 (slums, low-income areas, middle income, and high income). This is reflected in its water rate cost. Surveys and data mobilization have been undertaken by AITKEN and OXFAM on behalf of the GVWC with support from DFID in the ‘Strategic Water Supply and Sanitation Framework Part 2 Improvement Plan - Volume 1 Plan.

Result and Discussion

Line no-129: In Figure 2, could you please show the water sources in different ways - e.g. improved/unimproved source; pipe connection/unpiped connection, etc?

Thank you for your suggestion. Figure 2 has been revised, now as Table 1, with the different water sources as suggested in lines 150 - 152.

Line no-138-141: What is the significance of this information in your research?

Thank you for your observation.  In this survey demographic, socio-economic and physical characteristics influence the water needs and consumption patterns of households. The water consumption data sourced and analysed will serve as input data to three-dimensional modelling of groundwater development to assess the water budget and explore alternative approaches to augment the required supply

Line no-144: Please complete this sentence, not understandable.

Thank you for your observation. The sentence has been completed in lines 167 to 169.  

Line no-154: The relation you showed in this sentence is obvious, no need to establish it. Could you please show it in different ways?

Thank you for your observation. This relationship can be explained with the 20 developed models, where the R2 represents the proportion of variation in how an independent variable impacts the dependent variable. This has been moved as supplementary materials.   

LIne no-165: " this survey’s targeted university students to respond on behalf of their households," should be mention in the method section. This group may have better income and better access to water than other residents. How can you claim it as a representative group from the community?

This has been moved to methods section now Lines 93 – 97, 115 - 116.

Line no- 179: This may be 'Total HH percentage' not 'Total % distance'.

Thank you for the correction. This has been corrected.

Line no-197: "93 litres per capita per day (l/p/d)". How you measured the total averaged amount of water used? How do you clarify the reliability? Please write it in the method section.

This has been moved to the methods section now Lines 94 – 100.

Line no-216 & 294: what is the meaning of the full sample? It may be a total sample or please describe it in the methods section.

Thank you for the observation. This is explained in the methods section line 100 to 103.

The survey was conducted in the rainy (245households) and dry season (153 households). The total sample is a combination of the two surveys (398).

Line no-258: In the sentence "The smallest household size (2 persons) have the highest water consumption per...." How can you justify small HH sizes have the highest per capita water consumption?

Thank you! Similar findings were observed in the studies conducted by (Butler and Memon, 2006), (Hussien, Memon and Savic, 2016) - W. A. Hussien, F. A. Memon, and D. A. Savic, “Assessing and Modelling the Influence of Household Characteristics on Per Capita Water Consumption,” Water Resour. Manag., vol. 30, no. 9, pp. 2931–2955, 2016. DOI 10.1007/s11269-016-1314-x.

Butler, D. and Memon, F. A. (2006) ‘Water Demand Management David Butler and Fayyaz Ali Memon Department of Civil & Environmental Engineering’, (May 2014), pp. 2014–2017.

Line no-305: What is the heading of the last column of Table 3.

Thank you. Heading is ‘Comparison from past studies’ corrected in line 359.

Line no-347: In the sentence " average per capita drinking consumption is 4 liters per day" - What about the ranges of data? Could you please mention the seasonal variation in Freetown?

Thank you for your observations. The seasonal variation per capita consumption for drinking in the rainy season (May to November) ranges from 1 to 6 litres and in the dry season (December to April) ranges from 1 to 5 litres.

Line no-420: letter 'h' is missing in 'hig'.

Thank you. Noted and has been corrected in line 482.  

Line no-423: On what basis, you calculated the TW average volume of water use? Any references for calculation? Please mention this in the methods section.

Thank you. The methodology is explained in detail and has been added as an appendix. This reference to Hussein et al.  

The methodology is explained in details and has been added as an appendix

Line no-436: Is this the same survey or a different survey for your comparison?

Thank you. This is the same survey comparing the findings of the seasonal variability of per capita water  use in lines 485 to 586. 

Line no-453: Reference 51 of the sentence - Please complete this sentence, not understandable.

Thank you. Noted and has been corrected in line 507 to 508. 

Line no-504: "who sites that.." maybe "WHO sites that..." or any other suggestion?

Thank you! This statement has been revised in lines 559 to 560.

Line no-510: "2.06 brt/p/d" - what is brt?? Please write its full form.

 Thank you, line 565 has been Correction ‘as bathroom tap use per day’

Conclusion

Line no-532: Your conclusion seems abstract, very long, and not clear. Please rewrite it

Thank you, the conclusion has been revised in lines 597 to 608,

 Reference

Line no-583: Could you please add DOI on the references if applicable e.g. Referernce-1 (DOI 10.3390/w10030321 and Reference-3 (DOI 10.1029/2012wr012398)?

Thank you. Doi: have been added where applicable.

Line no-585: Please mention the website or ISBN of reference 2.

Thank you. This reference has been replaced with newer articles.

The ISBN is WRC Report No 1536/1/06 ISBN 978-1-77005-480-6

NOVEMBER.

Please check your all references from 1-58.

Thank you. Reference is revised.

Round 2

Reviewer 1 Report

Please correct Fig 4 and 5.

Regarding Fig 4, please correct the display of the x-axis.

Regarding Fig 5, the width of the histogram is not correct.

You should use tick widths such as 10, 15, 20. The width seems arbitrary.

Author Response

Dear Reviewer,

Thank you for your valuable comments and suggestion. 

Manuscript reference number: water-1008138

Peer-Reviewed Round 2

Reviewer #1:

Please correct Fig 4 and 5.

Response:

Figures 4 (lines 227 – 241) and 5 (lines 264 – 278) have been responded to as requested.

Regarding Fig 4, please correct the display of the x-axis.

Response:

The x-axis has been rectified. Thank you for your observation

Regarding Fig 5, the width of the histogram is not correct.

You should use tick widths such as 10, 15, 20. The width seems arbitrary.

Response:

Thank you for your keen observation in Figure 5, a missing segment of the data previously omitted has been corrected.  Tick width used is 15.

Reviewer 2 Report

Good job on the revisions. This will be a nice addition to the literature. 

One problem is that it appears the References section/field was not "refreshed" in MS Word to change/eliminate references. For example, Fan et al. was still listed as Ref 2 in References, but [2] was deleted from the text, and the old reference [1] in Line 36 still appeared in the References despite your statement that it was changed. However, there are new references added later in the manuscript, so I'm not sure what happened. 

Author Response

Dear Reviewer #2, 

Thank you very much for your valuable time, comments, and suggestions which have highly improve the quality of this manuscript. 

Manuscript reference number: water-1008138

Peer Reviewed Round 2

Reviewer #2:

One problem is that it appears the References section/field was not "refreshed" in MS Word to change/eliminate references. For example, Fan et al. was still listed as Ref 2 in References, but [2] was deleted from the text, and the old reference [1] in Line 36 still appeared in the References despite your statement that it was changed. However, there are new references added later in the manuscript, so I'm not sure what happened. 

Response

Thank you. We appreciate your thorough review. Reference [1] in line 36 is retained. The reference section has been refreshed to reflect the accurate citing and manual input of Doi’s, authour’s name inserted (lines 637 – 813) and sorted. 

Reviewer 3 Report

Dear Authors, thank you very much for your efforts in making all the corrections and clarification to the previous comments. In my opinion, you can improve your methodology and conclusion section. Your conclusion is not impressive though you improve a lot after the comments of the first review. Could you please rewrite the conclusion better with the significance of the research you did?

Author Response

Dear Reviewer#3,

Thank you very much for your valuable, time, comments, and suggestions which have greatly improved the quality of this manuscript. 

Reviewer # 3

Peer-Reviewed Round 2

Dear Authors, thank you very much for your efforts in making all the corrections and clarification to the previous comments. In my opinion, you can improve your methodology and conclusion section. Your conclusion is not impressive though you improve a lot after the comments on the first review. Could you please rewrite the conclusion better with the significance of the research you did?

Response

Thank you for the nice compliment! The study conclusion has been re-written in lines 610 – 633.
